# Insights into Species Preservation: Cryobanking of Rabbit Somatic and Pluripotent Stem Cells

**DOI:** 10.3390/ijms21197285

**Published:** 2020-10-02

**Authors:** Lucie Gavin-Plagne, Florence Perold, Pierre Osteil, Sophie Voisin, Synara Cristina Moreira, Quitterie Combourieu, Véronique Saïdou, Magali Mure, Gérard Louis, Anne Baudot, Samuel Buff, Thierry Joly, Marielle Afanassieff

**Affiliations:** 1Univ Lyon, Université Claude Bernard Lyon 1, Inserm, INRAE, Stem Cell and Brain Research Institute U 1208, USC 1361, F-69500 Bron, France; lucie.gavinplagne@imv-technologies.com (L.G.-P.); florence.perold@inserm.fr (F.P.); posteil@cmri.org.au (P.O.); voisinsoph@gmail.com (S.V.); Synaracris@hotmail.com (S.C.M.); quitterie.combourieu@ac-academie.fr (Q.C.); veronique.saidou@outlook.fr (V.S.); magali.mure@hotmail.fr (M.M.); 2Univ Lyon, Université Claude Bernard Lyon 1, VetAgro Sup, UPSP ICE 2016.A104, F-69280 Marcy l’Etoile, France; samuel.buff@vetagro-sup.fr (S.B.); tjoly@isara.fr (T.J.); 3Univ Paris, Université Descartes Paris V, LVTS, Inserm UMRS 1148, F-75018 Paris, France; gerard.louis@parisdescartes.fr (G.L.); anne.baudot@parisdescartes.fr (A.B.); 4Univ Lyon, Université Claude Bernard Lyon 1, ISARA-Lyon, UPSP ICE 2016.A104, F-69007 Lyon, France

**Keywords:** cryobanking, pluripotent stem cell, somatic cell, synthetic medium, rabbit, dimethyl sulfoxide

## Abstract

Induced pluripotent stem cells (iPSCs) are obtained by genetically reprogramming adult somatic cells via the overexpression of specific pluripotent genes. The resulting cells possess the same differentiation properties as blastocyst-stage embryonic stem cells (ESCs) and can be used to produce new individuals by embryonic complementation, nuclear transfer cloning, or in vitro fertilization after differentiation into male or female gametes. Therefore, iPSCs are highly valuable for preserving biodiversity and, together with somatic cells, can enlarge the pool of reproductive samples for cryobanking. In this study, we subjected rabbit iPSCs (rbiPSCs) and rabbit ear tissues to several cryopreservation conditions with the aim of defining safe and non-toxic slow-freezing protocols. We compared a commercial synthetic medium (STEM ALPHA.CRYO3) with a biological medium based on fetal bovine serum (FBS) together with low (0–5%) and high (10%) concentrations of dimethyl sulfoxide (DMSO). Our data demonstrated the efficacy of a CRYO3-based medium containing 4% DMSO for the cryopreservation of skin tissues and rbiPSCs. Specifically, this medium provided similar or even better biological results than the commonly used freezing medium composed of FBS and 10% DMSO. The results of this study therefore represent an encouraging first step towards the use of iPSCs for species preservation.

## 1. Introduction

Agricultural biodiversity is indispensable to food security, sustainable development, and many vital ecosystem services and supplies. Biodiversity increases the resilience of production systems and livelihoods to shocks and stresses, including the effects of climate change [1]. For centuries, however, livestock producers have used increasingly powerful technologies for animal selection and breeding, which have led to the alteration of genetic resources and the threatened extinction of approximately a third of all domestic animal strains [2]. In 2012, this loss of livestock biodiversity led to the creation of a French national network intended to link the various centers of biological resources (CRBs) for domestic animals. This infrastructure, the CRB-Anim [3], is managed by the French National Research Institute for Agriculture, Food and Environment (INRAE) and aims to preserve the biodiversity of different mammals, birds, fishes, shellfish, and bees raised to supply the food industry.

One research axis of the CRB-Anim involves the development of a method for the cryopreservation and reproductive use of somatic tissues via the application of induced pluripotent stem cell (iPSC) technologies. Pluripotency is defined as the ability of a single cell to produce all cell types, including germ cells, in a developing fetus. In vivo, this property is possessed by the epiblast, which serves as the foundational tissue of the whole embryo. In mice, pluripotent embryonic stem cells (mESCs) can be harvested and propagated in vitro from the inner cell mass (ICM) cells of the blastocyst [4,5]. However, pluripotency can be also obtained via the genetic reprogramming of adult somatic cells such as cutaneous fibroblasts by the overexpression of specific pluripotent genes, including *Oct4*, *Klf4*, *Sox2*, and *c-Myc* [6]. The resulting mouse induced pluripotent stem cells (miPSCs) possess the same properties as mESCs and can colonize a host embryo and participate in the development of all tissues [7]. The chimeric mice resulting from this process can then transmit the genetic features of the original somatic cells to their offspring. MiPSCs can also be differentiated into male [8] or female [9] functional gametes in culture, and these gametes can then be used to produce embryos via in vitro fertilization. Finally, miPSCs can be used as nucleus donor cells for nuclear transfer cloning [10,11]. In summary, iPSCs are useful tools for the preservation of endangered wildlife species and domestic animals [12,13].

The rabbit is both a subject of agricultural interest and a relevant model of various human conditions, including cardiovascular diseases, hypertension, diabetes, ophthalmologic disorders, and bacterial or viral infections [14,15,16]. The rabbit is also a promising bioreactor for the preparation of biological drugs (e.g., recombinant proteins and vaccines) derived from serum or milk [17]. Rabbit pluripotent stem cell (rbPSC) lines were first generated in three Asian laboratories during the early 2000s [18,19,20,21]. Both rabbit embryonic stem cell (rbESC) and rabbit induced pluripotent stem cell (rbiPSC) lines exhibited the cardinal features of pluripotency, including the capacities for long-term self-renewal; differentiation into ectodermal, mesodermal, and endodermal derivatives; and teratoma formation after injection into immunocompromised mice. Like most mammal pluripotent stem cells (PSCs), rbPSCs require basic fibroblast growth factor (FGF2)- and activin/nodal/transforming growth factor ß1 (TGFß1)-mediated signaling for pluripotency maintenance; unlike the gold-standard mPSCs, however, the rabbit cells do not depend on leukemia inhibitory factor (LIF)-mediated signaling [22,23]. This characteristic of rbPSCs is associated with a higher level of instability in culture [24] and a lower level of resistance to single-cell dissociation and freezing [25]. Culture, but also freezing, can induce the selection of sub-populations of PSCs presenting mutations [26] or epigenetic modifications with long-term putative effects on cells and/or their derivatives [27,28]. Similarly, somatic reprogramming is regulated by epigenetic phenomena [29] that could be affected by the epigenetic status of the cells to be reprogrammed. Therefore, the use of iPSCs for species preservation requires the development of a safe, standardized, and xeno-free freezing protocol that can be used for both stem cell banking and for tissue biopsies that will be used in later cell reprogramming approaches.

Currently, cells and small tissues are most commonly preserved via controlled-rate cooling in the presence of serum and 10% dimethyl sulfoxide (DMSO) as a cryoprotectant [30,31]. This technique can be easily applied using a freezing container or controlled-rate freezer and is most often used by cell biologists lacking expertise in cryobiology. However, two main risks are associated with this method: a health-related risk due to the use of serum and/or animal-derived products in freezing media, and a toxicity risk due to the use of high concentrations of DMSO [32]. To avoid the risks associated with serum, home-made and commercial freezing media has been supplemented with natural macromolecules, such as soybean lecithin [33] and silk sericin [34], or synthetic macromolecules, such as liposomes [35], polysaccharides [36], and polyvinylpyrrolidones [37], with varying degrees of success depending on the cryopreserved cells. Previous studies have already demonstrated the effectiveness of one type of commercial chemically defined media, STEM ALPHA.CRYO3 (hereinafter referred to as CRYO3), in the cryopreservation of rabbit embryos [38,39]. CRYO3 is a patented medium that lacks serum, protein, and dextran and is manufactured according to good manufacturing practice, in accordance with directive 2001/83/EC. This medium is composed of a synthetic high molecular weight (> 106 D) hyaluronic acid, glucose, carbohydrates, amino acids, mineral salts, vitamins, and fatty acid esters in a buffer solution. CRYO3 was initially designed to replace serum in the freezing medium used clinically to preserve human somatic and adult stem cells [40,41,42,43]. However, it was also shown to be effective for the cryopreservation of bovine embryos [44] and ovine sperm [45]. 

This study aimed to evaluate the effects of a slow-freezing protocol using a CRYO3-based medium in combination with a reduced concentration of DMSO on both the derivation of fibroblasts from frozen skin biopsies and the pluripotency characteristics of rbPSCs. The efficacy of CRYO3 as a potential serum substitute was evaluated using two approaches: a biological analysis of several properties of skin fibroblasts and rbPSCs after a slow-freezing and thawing protocol, and a thermodynamic analysis based on differential scanning calorimetry (DSC). 

## 2. Results

### 2.1. Experimental Designs

Two freezing/thawing experiments were designed. The first experiment aimed to define the cryopreservation conditions required to obtain reprogrammable rabbit fibroblasts (rbFs) in rbiPSCs from somatic tissues (Figure 1A). The second experiment aimed to determine the appropriate freezing conditions for rbPSCs versus mESCs, which differ in their resistance to single-cell dissociation (Figure 1B). 

Regarding biodiversity preservation, useful somatic tissues must be readily available for sampling, even from field-raised animals and sick or dead individuals, and for cryopreservation without the need for prior cell isolation. Accordingly, a maximum number of biopsy cells should remain alive and be easily derivable after the tissue is frozen and thawed, and the process should not require the complex vitrification procedures needed to preserve tissue structures and vascularization [46,47]. Therefore, we selected ear biopsies that met these requirements and determined the ideal slow-freezing cryopreservation conditions by testing five parameters: (i) the eventual storage of samples at 4 °C for 48 h before cryobanking; (ii) the tissues dissected from the biopsies (skin versus internal cartilage); (iii) the sizes of the frozen tissue pieces (3 mm^2^ versus 1 cm^2^); (iv) the freezing medium (synthetic (CRYO3) versus biological (FBS)); and (v) the percentage of the cryoprotective agent (DMSO; 4% versus 10%). In this first experiment, rabbit ears were dissected directly at the time of harvesting or stored at 4 °C in phosphate-buffered saline (PBS) + 1% 10,000 U/L penicillin + 10,000 U/mL streptomycin + 29.2 mg/mL L-glutamine (PSG) for 48 h. Next, the ears were dissected to separate the skin from the internal cartilage. Both tissues were cut into small (3 mm^2^) or large (1 cm^2^) pieces, washed in PBS-PSG and frozen in cryovials to −80 °C in CRYO3 or FBS containing 4% or 10% DMSO in a controlled-rate freezer (Cryologic CL8800i) at a cooling rate of 1 °C/min. The cryovials were then stored in liquid nitrogen. After at least 2 weeks, the cryovials were thawed directly from liquid nitrogen in a 37 °C water bath until all ice crystals were melted. Ten milliliters of Dulbecco’s modified Eagle’s medium (DMEM) were then added dropwise to the tissue pieces. The suspension was centrifuged for 5 min at 300× *g* and 20 °C, and the tissue pieces were then transferred to culture dishes containing rbF culture medium. Upon reaching confluency, the primary cells were passaged with trypsin-EDTA, and the time between the two first passages was recorded. Two to four independent samples from two to four different rabbits were analyzed per freezing condition (Table 1). The cells were then frozen under standard conditions (i.e., 90% FBS + 10% DMSO in a freezing container), as these 61 rbF lines obtained under the 32 conditions tested could not all be analyzed in the same time. To account for this additional freezing step, two types of controls were performed on unfrozen skin and cartilage samples: freshly derived rbF lines and rbF lines frozen at P2 like all other rbF lines derived from frozen tissues (Table 1). After thawing, in order to determine the rbF ability to be used for iPSC production, the incorporation of propidium iodide (PI), culture during six passages and transduction with a Sendai viral vector encoding the green fluorescent protein (GFP) were evaluated to determine the cell survival rate, growth rate, and transduction rate, respectively (Figure 1A). 

Both rbPSCs and mESCs are generally frozen under standard conditions (i.e., 90% FBS or knockout serum replacement (KOSR) + 10% DMSO in a freezing container). However, we observed that the cell mortality at thawing and cell passages is much higher for rbPSCs than for mESCs. This difference is related to the intrinsic mechanism supporting pluripotency in these two species [24], and consequently to the lower resistance of rbPSCs to single-cell dissociation compared to mESCs. In order to improve the freezing conditions of rbPSCs, we analyzed three parameters: the freezing medium, the percentage of cryoprotectant, and the slow-freezing method. Therefore, the second experiment compared cryopreserved single-cell suspensions of rbESCs, rbiPSCs, and mESCs stored in three types of freezing media (FBS or knockout serum replacement (KOSR; cell type-dependent), CRYO3 or CryoStor^®^ CS10 (ready-to-use medium containing 10% DMSO (Stemcell Technologies, Grenoble, France); positive control), six concentrations of DMSO (0%, 2.5%, 3%, 4%, 5%, or 10%), and two slow-freezing methods (freezing container or controlled-rate freezer). Three independent replicates were performed per condition. After thawing, the viability at passages 0 (P0, corresponding to thawing) and 5 (P5), growth recovery from P0 to P6 and gene expression at P1 and P6 were assessed. The freezing media were evaluated using a DSC-based thermodynamic approach (Figure 1B). 

### 2.2. Derivation of rbfs from Frozen Tissue

Nearly all tissue freezing conditions enabled the derivation of rbF lines (Table 1); however, these cells could not be derived from large pieces of frozen skin or cartilage isolated from ear biopsies stored at 4 °C for 48 h. The morphologies of the derived cells could be classified into four categories independently of their cutaneous or cartilaginous origins and the biopsy freezing conditions (Figure 2A, Appendix A): short fibroblasts, elongated fibroblasts, epithelial-type cells, or a mixture of these morphologies. No differences were observed in the times required to obtain rbF lines between the frozen tissues. However, these tissues required with significantly longer rbF derivation times, (Figure 2B), which ranged from 18 to 28 days for frozen skin samples versus 16 days for fresh skin controls and from 17 to 25 days for frozen cartilage samples versus 14 days for fresh cartilage controls (Appendix A). These variations in rbF derivation times were mainly due to the speed at which primary cells from the tissue pieces emerged and grew to produce confluent cultures in 100-mm dishes (Appendix A). After the first passages of these primary cultures, the rbF growth speeds were normalized between frozen and fresh tissue cells, with an average interval of 4.32 ± 1.76 days (Appendix A).

### 2.3. Quality of Frozen Tissue-Derived rbFs

After two derivation passages, all rbF lines were frozen according to the standard protocol for fibroblasts (i.e., a half-volume of FBS + 20% DMSO per half-volume of cell suspension in fibroblast medium) (Appendix A). Subsequently, most rbF lines were thawed, and the viability rates, growth rates, and GFP-expressing Sendai virus infectivity were analyzed. The latter analysis was selected as a criterion for iPSC reprogramming because primate iPSC lines are often generated using non-integrated Sendai vectors. 

Immediately after thawing, the viability rates of rbF lines ranged from 68.7% to 99.2% and no difference was observed between skin- (in green) or cartilage- (in pink) derived rbF lines for both samples and control (Figure 3Aa and Appendix A). We therefore compared the viability of rbF derived from the 12 freezing conditions used (Figure 3Ab); only four of them showed significant differences (rbF derived from tissue frozen in FBS/10%/large/0 h versus those derived from tissue frozen in FBS/10%/small/48 h, FBS/4%/small/48 h or CRYO3/10%/small/48 h). According to the composition of these four freezing conditions, the common parameter that seems to be responsible for this difference is the size of the frozen fragments. Nevertheless, very few samples were included in these four freezing conditions, which may bias the statistical test. Viability rates were more likely to be directly related to the qualities of the rbF freeze–thaw procedure rather than to the freezing conditions of the original tissues.

The rbF growth curves of lines derived from frozen tissues were fairly heterogeneous (Figure 3B and Appendix A), but the slopes of the curves only differed significantly between samples frozen with 4% and 10% DMSO. No statistical difference in rbF growth were induced by the original medium or tissue used for freezing. However, the mean slope of the rbF sample curves was statistically lower than those of the fresh and frozen rbF controls, suggesting that cells from frozen tissues tend to grow more slowly than cells from fresh tissue. This observation indicates that the tissue freezing step may select more resistant but slower growing cells or may alter the growth mechanisms of resistant cells via epigenetic modifications. Similarly, the mean slope of the fresh rbF controls was statistically higher than that of frozen rbF controls, again showing an effect of the freezing procedure on cell growth.

The infection rates of rbFs from frozen tissues ranged from 19.9% to 94.4% (Figure 3), and no significant correlations were observed between the infection rate and the tissue origin or freezing condition, except for the size of tissue fragments (Appendix A). Surprisingly, the infection rates of fresh or frozen rbF controls were much lower (23.4 ± 13.4% and 20.4 ± 8%, respectively) than those of rbFs derived from CRYO3- or FBS-frozen tissues (67 ± 21.9% and 67.4 ± 19%, respectively) (*p* < 0.001). This highly significant difference may be attributable to three additive facts: a selection of more receptive cells by the freezing step of the tissue, a massive loss of infectivity of the Sendai virus aliquots used between experiments with rbF sample and control lines, and variable susceptibilities of the animals or tissues to the virus (Appendix A). Based on our previous experiences (data not shown), the average rates of rbF infection with the Sendai virus ranged from 65% to 70% and were consistent with the results obtained with frozen tissue-derived rbF lines, depending on the receptivity of the tissues or susceptibility of the animals (Appendix A). Accordingly, there was likely an issue related to Sendai virus infection in the control rbF lines.

Overall, these data showed that the percentage of DMSO and the size of tissue fragments have a substantial effect on the quality of frozen tissue-derived rbFs. The original tissue, the medium, and the storage time of the tissue before freezing have no significant impact on any of the analyzed factors. Finally, the best conditions for freezing ear tissue to obtain reprogrammable rbFs are treatment of small pieces of skin or cartilage, with or without prior storage of the ears for 48 h in PBS at 4°C, in a medium based on CRYO3 or FBS with 10% DMSO, using a controlled-rate freezer. 

### 2.4. Viability of Frozen–Thawed PSCs

Before freezing, the viability rates of mESCs, rbESCs, and rbiPSCs were 92.33 ± 2.45%, 88.17 ± 1.52%, and 92.93 ± 1.27%, respectively. After thawing, the viability rates of all three cell types decreased significantly as the percentage of DMSO decreased, particularly when the percentage of DMSO was 4% or less, regardless whether FBS- or CRYO3-based freezing media had been used (Figure 4). For mESCs and rbiPSCs, significant differences in viability were also observed between cells frozen in a freezing container (Figure 4A) and the controlled-rate freezer (Figure 4B); the former was associated with less variability and better viability regardless of the concentration of DMSO. Larger error bars were observed for the controlled-rate freezer (Figure 4B) compared with the freezing container (Figure 4A). A difference in the cooling curves used in the two methods might explain these larger error bars. The cooling curve was linear in the controlled-rate freezer while the cooling curve slowed down considerably at the end of the cooling cycle with the freezing container. The cooling curve of the controlled-rate freezer might not be appropriate or not similar enough to that of the freezing container in the −80 °C freezer. A second explanation is that the cell lines used for this study are usually frozen in a freezing container and the cells are somehow adapted to this method. On the contrary, the cells are not adapted to freezing by the controlled-rate freezer method which would then induce more cellular stress.

Similar rates of viability were observed for mESCs frozen in media containing 10% DMSO, CryoStor^®^ CS10 (86.43 ± 1.59%) and fresh cells (92.33 ± 2.45%) when a freezing container was used (Figure 4Aa). Lower viability rates were achieved with a controlled-rate freezer than with a freezing container under most conditions, even for positive control cells (cells frozen in CryoStor^®^ CS10; 84.53 ± 0.58%) (Figure 4Ba). However, when using the controlled-rate freezer, no significant difference was observed for mESCs frozen in FBS- or CRYO3-based media containing 5% or 10% DMSO (Figure 4Ba). The interaction between the method and the percentage of DMSO was only significant at 5% of DMSO.

Most of the parameters had a significant effect on the viability of rbESCs and the interaction between the method and the medium was noteworthy (Figure 4b): CRYO3-based medium improved the viability of rbESCs frozen in a freezing container, while FBS-based medium improved the viability when a controlled-rate freezer was used. However, the viabilities observed with a freezing container using 10% DMSO were similar between the positive control (cells frozen in CryoStor^®^ CS10; 71.07 ± 2.08%) and samples frozen in both media (Figure 4Ab) and lower than those of fresh cells (88.17 ± 1.52%), indicating sub-optimal freezing conditions for the rbESCs, even with the freezing container. 

No significant difference was observed between rbiPSCs frozen in CRYO3- or KOSR-based media containing 4%, 5%, and 10% DMSO with both methods (Figure 4c). In addition, the viability rates of the latter conditions did not significantly differ from those of fresh sample (92.93 ± 1.27%) and positive control (cells frozen in CryoStor^®^ CS10; 89.60 ± 0.17%) when using a freezing container (Figure 4Ac). The interaction between the method and the percentage of DMSO was also highly significant, confirming the sub-optimal freezing conditions used with the controller-rate freezer.

Overall, these results are consistent with the ability of both synthetic media, CRYO3 and CryoStor CS10, to preserve the viability of mouse and rabbit PSCs after a freeze-thaw cycle in a freezer container. In addition, the level of DMSO in the CRYO3-based medium can be reduced without affecting cell viability to 5% or 4%. These data also show that the decreasing temperature curve must be better suited to efficiently use a rate-controlled freezer. Nevertheless, after five cell culture passages, the viability rates of all cell types were similar to those of the fresh cells, regardless the initial freezing conditions (Appendix A), showing no effect of freezing conditions on the viability of recovered cell.

### 2.5. Growth Curves of Frozen–Thawed PSCs

Compared to fresh cells, frozen rabbit PSCs required a recovery period of at least 5 days before reaching the normal growth rate, regardless of the freezing medium, freezing method and DMSO concentration. This recovery period was also observed with the control cells frozen in the CryoStor^®^ CS10 (Figure 5b,c). In addition, under certain freezing conditions (in FBS/KOSR-based media and CryoStor^®^ CS10), rbPSC growth curves showed an inflection during the first five days of culture, which is possibly due to a higher rate of cell death, resulting in a longer recovery period. In contrast, frozen mouse cells recovered rapidly and grew at their usual rate (Figure 5a). This difference in recovery between mouse and rabbit PSCs may be due to the intrinsic stability, better resistance to cell dissociation and higher viability of mPSCs relative to rbPSCs.

Only time, freezing method, and DMSO concentration had significant effects on the total numbers of recovered mESCs (Figure 5a). No significant difference was noticed between curve slopes of fresh cells, control cells and sample cells frozen in FBS- or CRYO3-based media in a freezing container, particularly with high concentration of DMSO (Figure 5Aa and Appendix A). In the same way, similar curve slopes were observed between mESCs frozen in FBS- or CRYO3-based media in a controlled-rate freezer, regardless the concentration of DMSO (Figure 5Aa and Appendix A). However, the latter conditions resulted in slower growth of the frozen mESCs than that of fresh or control cells.

For rbPSCs, all the fixed parameters (e.g., freezing method, freezing medium, and DMSO concentration), as well as time, significantly influenced the total numbers of cells recovered (Figure 5b,c). Irrespective of method and DMSO concentration, CRYO3-based medium showed significantly increased growth of rbiPSCs and rbESCs than FBS/KOSR-based medium and CryoStor^®^ CS10 control medium. In addition, no significant difference was observed in the total rbiPSCs recovered after freezing in 3%, 4%, 5%, and 10% of DMSO (Figure 5C and Appendix A), regardless of the medium and method used, suggesting that rabbit cells could be cryopreserved with low concentration of cryoprotectant. The same trend seems to be true for the rbESCs frozen with 5% of DMSO (Figure 5b and Appendix A), but should be confirmed with more optimal freezing conditions, particularly with the controlled rate freezer. Once rabbit cells had fully recovered, the slopes of the linear portion of the growth curves were similar within each cell type frozen in a defined medium, regardless of the concentration of DMSO (Appendix A). However, as previously described for rabbit cells, the CRYO3-based medium induced less cell death and allowed faster recovery during the first few days of culture (Figure 5b,c). In addition, compared to FBS/KOSR-based media, a lower fluctuation in cell growth rates between replicates was observed with CRYO3-based medium (Appendix A).

In conclusion, the relative fragility of rbPSCs compared to mESCs has influenced the recovery and growth of the frozen cells under all freezing conditions, indicating the need to improve their cryopreservation methods. However, the CRYO3-based medium and the low percentage of cryoprotectant (4–5% of DMSO) appeared to increase the growth rate of rabbit cells compared to the FBS/KOSR-based media usually used with 10% DMSO.

### 2.6. Gene Expression in Frozen–Thawed PSCs

We next analyzed the expression of three pluripotency network genes (*OCT4*, *NANOG*, and *ESRRB*) and three pluripotency marker genes, including two genes specific for mESCs (*Rex1* and *Cdh1*) and one specific for rbPSCs (*CDH2*) (Figure 6 and Appendix A). The expression data were normalized to the expression of the same genes in fresh cells; here, an expression value closer to zero indicates a level of expression more similar to that in fresh cells. We observed more heterogeneous gene expression in cells that had been frozen in a controlled-rate freezer (Appendix A), compared to cells frozen in a freezing container (Figure 6), regardless of the cell type and freezing medium. Moreover, this heterogeneity was reduced in cells analyzed at passage 6 relative to those analyzed at passage 1, indicating that the expression of pluripotency genes had stabilized once the cells had fully recovered and began to grow normally. Generally, when a freezing container was used, cells frozen in CRYO3-based medium exhibited gene expression patterns more similar to those of fresh cells (Figure 6). In contrast, when a controlled-rate freezer was used, cells frozen in FBS/KOSR-based media exhibited expression patterns more similar to those of fresh cells (Appendix A). 

For mESCs, we mainly observed significant effects of the freezing method on expression of *Oct4*, *Nanog*, and *Rex1* genes. The linear model used did not reveal any significant difference in *Esrrb* expression (Figure 6). For both rbESCs and rbiPSCs, the main factor showing a significant effect on the expression of most of the six genes analyzed (*OCT4*, *NANOG*, *REX1*, *ESRRB*, *CDH1*, and *CDH2*) was the passages. The freezing medium and method also induced a significant individual effect on the expression of all five genes, with the exception of *REX1*, in rbiPSCs. In contrast, the expression of *REX1*, like that of *CDH2*, was significantly influenced by the freezing method in rbESCs. In the latter, the freezing medium induced significant variations in the expression of *CDH1* and *CDH2*. Surprisingly, the percentage of DMSO itself did not generate significant variations in gene expression, with the exception of *Nanog* in mESCs and *CDH2* in rbESCs. However, we observed that at passage 6, the gene expression patterns of samples frozen with 3%, 4%, or 5% DMSO were generally more similar to those of fresh samples than to those of samples frozen with other DMSO concentrations, especially with the traditional 10% concentration. More importantly, gene expressions obtained with control cells frozen in CryoStor^®^ CS10 were not significantly closer to those of fresh cells and in some cases even lower than those of rbPSCS frozen in FBS/KOSR- and CRYO3-based media containing 10% DMSO.

Overall, our data showed that PSC freezing conditions influenced the expression of pluripotency network and marker genes after a few days in culture. The more the freezing conditions were adapted to the cells, the closer the expression levels were to those of fresh cells, as shown by the mESC expression results. However, after the recovery of the cells for five passages in culture, gene expressions tented to stabilize at the same levels than those of fresh cells, especially in cells frozen with 3–5% DMSO in CRYO3-based medium using a freezing container or in FBS/KOSR-based medium using a controlled-rate freezer.

### 2.7. Thermodynamic Properties of Freezing Media

Predictably, both the melting temperature (T_m_) and the maximum crystallization enthalpy (∆H) decreased as the DMSO concentration increased in all tested types of freezing media (Table 2). However, a similar pattern was not observed for the crystallization temperature (T_c_). These data suggest that as the concentration of DMSO in the freezing medium increases, the phase equilibrium temperatures decrease and a reduced amount of ice is formed. Cryostor^®^ CS10 yielded the lowest T_c_, T_m_, and ∆H. Paradoxically, CRYO3-based media were less thermodynamically stable, with high measured ∆H values, in the presence of 5% or 10% DMSO, compared to media containing FBS or KOSR.

## 3. Discussion

This study aimed to define safe and non-toxic cryopreservation protocols for skin biopsy samples and PSCs intended for the preservation of biodiversity. We chose to use the rabbit, as it is the only agronomic mammal from which ESCs and iPSCs have been derived and extensively analyzed [48]. As noted above, the potential risks of pathogen exposure and variable freezing processes associated with animal products have highlighted a need to replace these products with synthetic and chemically defined molecules for cryopreservation. Moreover, cryoprotectants such as DMSO confer a risk of toxicity when used at high concentrations (e.g., 10%). Consequently, we compared the effects of synthetic and biological media containing low or high percentages of DMSO on the cryopreservation of rabbit skin samples and rbPSCs, using a slow-freezing protocol with a cooling rate of 1 °C/min. All data from this study indicate the efficacy of a medium composed of CRYO3 and 4% DMSO for tissue and rbPSC cryopreservation, which was shown to yield biological results similar to than those associated with the traditional freezing medium composed of FBS and 10% DMSO [18,30,49].

With respect to biodiversity preservation, as described above, we selected ear biopsies for their ease of sampling under most breeding conditions and their ability to be frozen with a minimum of processing. We determined the best slow-freezing cryopreservation parameters by comparing five conditions: (i) immediate freezing versus initial storage of biopsies, (ii) skin versus internal cartilage dissected from the biopsies, (iii) small versus large pieces of frozen tissue, (iv) synthetic versus biological freezing medium, and (v) low versus high percentage of DMSO. All conditions were tested using two to four separate samples from two to four different rabbits, and the times required to derive fibroblasts were compared between frozen tissues and primary cultures obtained from fresh biopsies. Primary cells were produced from thawed tissues subjected to most freezing conditions, with the exception of large pieces of skin or cartilage issued from biopsies stored at 4 °C for 48 h. The latter result could be explained by a loss of cell viability before the freezing step and might be improved by storage in culture medium instead of PBS, as most vessel or cartilage biopsies used for homografting are stored at 4 °C for 1–3 weeks in a suitable medium that is refreshed every 2–3 days [50,51]. These conditions were shown to reduce the survival rates of chondrocytes in cartilage biopsies from 90% to only 78% between the 7th and 35th days of storage [51]. 

We further observed small variations in the times required to obtain confluent primary cultures that were likely associated with cell survival after tissue thawing, as demonstrated by the statistically significant differences in this parameter between fresh and frozen samples. However, the observed variations in cell morphology could not be attributed to any of the analyzed parameters and may also depend on the proportion of each viable cell type in a thawed biopsy tissue. After derivation, all rbF lines exhibited similar viability, growth, and Sendai vector transduction rates, regardless of the freezing condition, initial origin of the primary tissue, or morphology of the derived cells. Only two conditions, the size of the frozen fragments and the percentage of DMSO, influenced the viability, and transduction rates of the rbF lines and the growth of thawed cells, respectively. However, it should be noticed that the low number of samples included in some analyzed groups may have biased the statistical tests. Finally, all these data reveal that an all-or-nothing event results in cutaneous rbF lines with slower growth rates but features that are otherwise identical to those of freshly derived lines. All these data are consistent with the ability to reprogram rbFs derived from frozen and fresh samples into rbiPSCs. Therefore, from the perspective of health and toxicity, it seems entirely feasible to apply the safest cryopreservation protocol, which comprises the synthetic medium CRYO3 and 4% DMSO, to skin biopsies.

Regarding the cryopreservation of rbPSCs, we demonstrated that CRYO3-based medium could replace FBS- or KOSR-based media, especially if the concentration of DMSO is reduced to 4–5%. This synthetic medium induced acceptable viability rates, growth recoveries and pluripotency gene expression when supplemented with 4–5% cryoprotectant. These data are consistent with our previous studies on embryo cryopreservation [38,39,44] and with other studies which showed that FBS could be replaced by a commercial synthetic media containing a high molecular weight polymer when freezing human or mouse PSCs [52,53,54,55]. In our study, we compared CRYO3, a synthetic commercial product accredited for clinical purposes, with CryoStor^®^ CS10, a widely used stem cell freezing medium of unknown composition. CRYO3 is composed of hyaluronic acid, a molecule largely used in tissue engineering [56,57,58] and cryopreservation applications [59,60,61]. At a set DMSO concentration of 10%, mouse and rabbit PSCs frozen in CryoStor^®^ CS10 had better viability rates after thawing than the cells frozen in CRYO3; moreover, a DSC analysis demonstrated that the former condition yielded the lowest Tm and ∆H. Nevertheless, this medium did not improve the recovery of stem cell growth or better preserve the cell quality, as demonstrated by the levels of pluripotency gene expression after the thawed rbPSCs had recovered. It would be interesting to compare both synthetic media in the presence of 5% DMSO, as CRYO3 yielded similar or even better overall results at a lower cryoprotectant percentage. Moreover, the risk of toxicity increases as the cryoprotectant concentration increases. However, CryoStor^®^ CS5 was not available at the time of these experiments. 

Previous studies have demonstrated increased cell differentiation and decreased *OCT4* expression in human PSCs frozen in the presence of DMSO [62,63]. We did not observe such effects of DMSO concentrations on gene expression and cell differentiation. Clinically, DMSO residues can trigger adverse effects such as nausea, headache, and diarrhea [64,65,66,67]. In our study, we observed similar results between samples frozen in CRYO3-based media containing 4%, 5%, and 10% DMSO, indicating that it is possible to decrease the concentration of DMSO while maintaining a high level of conservation. A previous study demonstrated equivalent viability rates in mPSCs frozen in another synthetic medium (HypoThermosol FRS, Bio-Life Solutions) supplemented with 2% or 5% DMSO, compared to 10% DMSO [53]. In that study, however, after 2 days of culture, the viability rates decreased to 10% and 40% for cells frozen in 4% and 5% DMSO, respectively. In contrast, we observed an improvement in the quality of the recovered cells over time. Many researchers have reported the substitution or combination of DMSO with other cryoprotectants, such as ethylene glycol or polyethylene glycol [55,63,68,69,70,71]. However, these cryoprotectants were shown to be more genotoxic than DMSO when cryopreserving human oocytes [72]. In the same way, a vitrification method was proposed for the cryopreservation of PSCs, and particularly human PSC clumps that are sensitive to cell dissociation [25,55]. However, vitrification requires higher concentrations of cryoprotectants and may significantly increase the risk of cytotoxicity [32]. In our study, all the rbPSC lines were resistant to single cell dissociation and therefore a slow-freezing procedure was appropriate. 

We further demonstrated that rbPSCs could be frozen either in a controlled-rate freezer or in a freezing container. Surprisingly, the use of a freezing container induced better and less variable survival rates after thawing, whereas a previous study demonstrated that the use of a controlled-rate freezer improved the viability of rhesus macaque PSCs by 60–70% [69]. As mentioned before, this result can be explained by the previous adaptation of PSCs to the freezing containers or by the suboptimal conditions used with the controlled-rate freezer. In slow-freezing protocols, crystallization control is crucial to avoid the formation of heterogenous and chaotic crystals. This control could be improved by manual nucleation induction or seeding [73]. A seeding step before using the controlled-rate freezer has been shown to improve PSC freezing [30,49], although important variations were also observed with human PSCs using seeding at −7 °C or −10 °C [74,75,76].

The thermodynamic characterization of our freezing media was consistent with the knowledge in cryobiology about physical properties of cryoprotective solutions and the quality analysis of the recovered cells. Indeed, the T_m_ and ΔH decreased as the percentage of DMSO increased. However, an inverse relationship was not observed between the DMSO percentage and the T_c_ because the latter parameter depends more on the experimental conditions than on the sample composition. In fact, crystallization via heterogeneous nucleation may be induced by the presence of impurities on the surface of the aluminum pan, resulting in a variable T_c_. The thermodynamic stability of CryoStor^®^ CS10 suggested the formation of less ice at lower temperatures, which was consistent with the high rates of viability in thawed mESCs and rbPSCs that had been frozen in this medium. We further observed that the CRYO3-based medium exhibited a more variable ∆H than CryoStor^®^ CS10, suggesting that the former was less thermodynamically stable. However, this instability of CRYO3 was not previously observed in studies of embryo freezing media [38,44]. Except for CryoStor^®^ CS10, all freezing media exhibited similar thermodynamic patterns that corresponded to reduced viability in the thawed cells. These data demonstrate the interest of a DSC-based thermodynamic approach when analyzing the physical behaviors of freezing media and establishing appropriate freezing protocols. 

In conclusion, this study indicates the efficacy of a medium composed of CRYO3 and 4% DMSO for the cryopreservation of tissues and rbPSCs, and represents an encouraging first step towards the use of rbiPSCs for species preservation. However, although this technique appears to be successful in preserving the quality of fibroblasts and iPSCs, the genetic and epigenetic heritage of the frozen cells needs be tested and the differentiation capabilities of frozen iPSCs needs to be verified to ensure a complete preservation. Moreover, the efficacy of the freezing method can only be demonstrated by comparing the reprogramming into rbiPSCs of different rbF lines obtained from frozen and fresh tissues. Similarly, the ability of the rbiPSC lines thus obtained and in turn cryopreserved to be used to produce a new generation of rabbits should also be studied. However, at present, the techniques envisaged for species preservation—i.e., embryo colonization, gamete differentiation, or nuclear transfer cloning—are fully effective only in mice [13]. Much research therefore remains to be done to achieve this goal in rabbits as well as in other farmed and wild mammals.

## 4. Materials and Methods

### 4.1. Animals

New Zealand White rabbits were purchased from HyPharm (Roussay, France) and housed in a conventional animal care facility in compliance with European animal welfare rules (European parliament directive 2010/63/UE). The ear biopsy protocol was approved by the Ethics Committee for Experimental Neurosciences of Lyon (CELYNE, French committee #42) (20 April 2017) and registered by the French Ministry of Higher Education, Research and Innovation under animal experimentation project number APAFIS#6438-2016081120471028v6, 01 Dec 2017).

### 4.2. Cell Culture

Rabbit fibroblasts (rbFs) were derived from fresh or frozen ear skin biopsies. The ears of euthanized adult rabbits were dissected to separate the skin from the internal cartilage. Both tissues were cut into small (3 mm^2^) pieces and cultured on 0.2% gelatin-treated plates in DMEM supplemented with 20% FBS (Hyclone, Thermo Fisher Scientific, Waltham, MA, USA) + 1% of a solution containing PSG + 100mM ß-mercaptoethanol (ß-Met) + 1% non-essential amino acids (NEAA). The rbPSCs were cultured on mouse embryonic fibroblast (MEFs) feeder cells that had been prepared from 12.5-day-old embryos of the CF1 strain (Charles River, Saint-Germain-Nuelles, France). The MEFs were prepared, cultured, and treated with mitomycin-C (Sigma-Aldrich, Lyon, France) as described elsewhere [77,78]. RbESCs (line AKSL20) [79] were maintained at 38 °C in an atmosphere of 5% CO_2_ + 5% O_2_ and cultured on mitomycin C-treated MEFs (1.25 × 10^4^ MEFs/cm^2^) in DMEM/F12 medium supplemented with 10% KOSR and 10% FBS, 1% NEAA, 1% PSG, 1 mM sodium pyruvate (NaPy), 100 mM ß-Met, and 10,000 U of LIF. The medium was refreshed every 24 h. The rbESCs were dissociated every 2 days into single-cell suspensions after treatment with 1 × StemPro^®^ Accutase (Sigma-Aldrich, Lyon, France) and seeded onto 6-well plates at a density of 0.4 × 10^6^ cells/well. RbiPSCs (line B19) [80] were maintained under the same conditions as rbESCs, except that the medium contained 20% KOSR without FBS, and 10 ng/mL fibroblast growth factor (FGF2) in place of LIF. MESCs (line E14Tg2a) [81] were cultured on gelatin in Glasgow minimum essential medium (GMEM)–BRK supplemented with 10% FBS, 1% NEAA, 1% PSG, 1 mM NaPy, 100 mM ß-Met, and 10,000 U of LIF at 37 °C and 5% CO_2_. The media of rbiPSCs and mESCs were refreshed every 24 h. The cells were routinely dissociated into single-cell suspensions after treatment with 0.05% trypsin–EDTA and seeded onto 6-well plates at a density of 0.4 × 10^6^ cells/well or 1.5 × 10^5^ cells/well, respectively. All medium components were purchased from Invitrogen (Villebon-sur-Yvette, France) unless otherwise specified.

### 4.3. Cryopreservation

All tissue pieces were frozen in the selected freezing medium as described in the Experimental Design section, stored in liquid nitrogen and thawed directly at 37 °C in a water bath until the last ice crystals melted. The tissue pieces were then diluted by the dropwise addition of 10 mL of DMEM. The suspensions were centrifuged for 5 min at 300 g and transferred to culture dishes containing rbF culture medium. After deriving the rbF lines, the cells were frozen in conventional medium consisting of 90% FBS + 10% DMSO (Sigma-Aldrich, Lyon, France), and thawed in the same manner as described below for PSCs. All freezing media were prepared the day before use by mixing FBS, KOSR or CRYO3 (STEM ALPHA.CRYO3, Stem Alpha, Saint-Genis-l’Argentière, France) with 0%, 5%, 6%, 8%, 10%, or 20% DMSO (Appendix A) and were stored at 4 °C. After dissociation and centrifugation (300× *g* for 5 min), the cells were re-suspended either in culture medium (FBS and KOSR conditions), CRYO3 without DMSO or CryoStor^®^ CS10 (Stemcell Technologies, Grenoble, France). For FBS, KOSR and CRYO3 conditions, the cell suspensions were split into 0.5-mL aliquots in cryotubes, to which equal volumes of freezing medium at 4 °C were added dropwise. The following final cell concentrations were achieved: 2.5 × 10^6^ cells/mL for mESCs and 2 × 10^6^ cells/mL for rbESCs and rbiPSCs. The cryotubes were cooled to 4 °C prior to slow freezing. 

The cryotubes were transferred either to a freezing container (NALGENE^TM^ Cryo 1 °C Freezing Container, Sigma Aldrich, Lyon, France) placed in a −80 °C freezer for 15 h or to a controlled-rate freezer (Cryologic CL 8800i, Cryologic, Blackburn, Australia) with a set cooling rate of 1 °C/min with a limit of −80 °C. The frozen cryovials were then plunged into liquid nitrogen for storage. The cooling rate within the cryotube in the presence of cells and freezing medium was calculated using a thermocouple type “T” (TC Direct, Dardilly, France). The approximate cooling rates in the linear parts of the curves were 0.6 °C/min and 0.8 °C/min for the freezing container and controlled-rate freezer, respectively. The cryotubes were stored for at least 2 weeks in liquid nitrogen prior to thawing for assessment. 

The cryotubes were thawed directly from the liquid nitrogen in a water bath at 37 °C until the last ice crystals had melted. The cells were then diluted first by the dropwise addition of 1 mL of 37 °C culture medium and transferred to a 15-mL tube. The cryovial was then rinsed with 1 mL of 37 °C culture medium, which was transferred to the 15-mL tube. Finally, 4 mL of culture medium at 37 °C were added dropwise to the thawed cells. The suspension was centrifuged (300× *g* for 5 min), the supernatant was removed and the cells were resuspended in 1 mL of culture medium. A volume equal to a fifth of the thawed cell suspension volume was subjected to a flow cytometry assessment, while the remainder was cultured in a 6-well plate. The day after thawing, the cells were rinsed twice with PBS to remove dead cells. 

### 4.4. Flow Cytometry Assessment of Viability

Cell viability was assessed after thawing (P0) or after dissociation (P6 or fresh samples) by adding 5 μL of propidium iodide (1 mg/mL, Sigma Aldrich) to 200 μL of cell suspension 1 min before the analysis. The analyses were performed on a FACSCanto II cytometer (Becton Dickinson, Le Pont de Claix, France) and acquired using the FACSDiva software V9.0 (Becton Dickinson, San Jose, CA, USA). The fluorescence was evaluated using a 488-nm excitation laser and a 650 ± 13-nm bandpass emission filter.

### 4.5. Growth Curves

Growth recovery was calculated by counting the cells (Countess^TM^ II FL, Thermo Fisher Scientific, Villebon-sur-Yvette, France) at each passage from the first plating after cell thawing to the sixth cell passage (P0–P6). The results are presented as the logarithmic values of the total numbers of counted cells.

### 4.6. Infection Rate

The ability of rbFs to produce iPSCs was evaluated at P4. This parameter was determined as the rate of cellular infection resulting from transduction with the GFP-expressing Sendai viral vector. Triplicates of 5 × 10^4^ cells were plated on well of a 12-well plate in the presence of 250 µL of rbF medium and 5 µL of CytoTune-EmGFP Sendai Fluorescence Reporter (Life Technologies, Villebon-sur-Yvette, France) at a concentration of 1 × 10^6^ pi/mL (multiplicity of infection: 5). Each plate was centrifuged (570× *g* for 1 h) and incubated overnight at 37 °C and 5% CO_2_. One milliliter of rbF medium was added to each well, and the cells were cultured for 3 days before dissociation and analysis on a FACSCanto II cytometer equipped with FACSDiva software. The fluorescence was evaluated using a 488-nm excitation laser and a 530 ± 30-nm bandpass emission filter. 

### 4.7. Gene Expression

Total RNA was isolated from the cell pellets using QIAshredder, the RNase-Free DNase Set, and RNeasy Mini Kit (Qiagen, Les Ullis, France) according to the manufacturer’s instructions. The RNA was quantified using a NanoDrop 2000 spectrophotometer (Thermo Fisher Scientific, Villebon-sur-Yvette, France)). Next, 500 ng of RNA were reverse-transcribed using the High-Capacity RNA-to-cDNATM Kit (Thermo Fisher Scientific, Villebon-sur-Yvette, France)), according to the manufacturer’s instructions. Quantitative PCR (qPCR) was performed using the Fast SYBR^®^ Green Master Mix (Thermo Fisher Scientific, Villebon-sur-Yvette, France)), and all reactions were run on a StepOnePlusTM real-time PCR system (Applied Biosystems, Foster City, CA, USA) for 40 amplification cycles. A melt-curve analysis was used to verify that only the desired PCR products had been amplified. The qPCR efficiencies for both the target and reference genes were determined from the relative quantitative values for normalized target (calibrator) gene expression and calculated using StepOnePlus Software V2.1 (Applied Biosystems, Foster City, CA, USA). In all cases, the expression of the target genes was normalized to that of the rabbit TATA-box binding protein (*TBP*) gene or mouse β-actin (*Actb*). The expression of target genes in the frozen samples was also normalized to that in fresh samples (reference samples), and the data were quantified using the ΔΔCT method. All qPCR assays were performed in triplicate and repeated in at least three independent experiments. The results are presented as the logarithmic fold change values. All primers used for qPCR are shown in Appendix A.

### 4.8. Differential Scanning Clorimetry

The following media were analyzed by DSC: mESC, rbESC, and rbiPSC freezing media containing 0%, 5%, or 10% DMSO, and CryoStor^®^ CS10. The phase transitions of the freezing media were characterized using a power compensation DSC (Diamond DSC, Perkin-Elmer, Waltham, MA, USA) equipped with a liquid nitrogen cooling accessory (Cryofill) and Pyris software V11.11.1 (Perkin-Elmer, Waltham, MA, USA). The technical specifications of this DSC provided accuracy ranges of ± 0.11 °C for temperature and ± 1.10% for energy. The DSC was calibrated for temperature and energy at +2.5 °C/min using two standards: the melting of pure bi-osmosed water ice (0 °C; 333.40 J/g) and the crystallographic transition of solid-state cyclohexane (−87.06 °C; 79.58 J/g), which yielded a high data range of 720 mW. 

The DSC experiments were conducted using standard hermetically sealed aluminum pans (Perkin-Elmer, Waltham, MA, USA) designed for volatile samples. The pans had been cleaned previously using the standard procedure provided by Perkin-Elmer. The pans were first weighed on a highly sensitive balance scale (XS105 DualRange, Mettler Toledo, Viroflay, France) without the cryopreservation solution, and were weighed again after the cryopreservation solution was loaded to determine the sample masses at a resolution of 10^−5^ g. The mean sample weight (±standard deviation) was 4.06 ± 0.59 mg (*n* = 39). For the DSC, an empty oven baseline was regularly and carefully recorded using the same protocol for recording temperature variations, and this value was systematically subtracted from the sample thermograms. 

Each sample was subjected to two cycles of cooling and warming between 10 °C and −150 °C to determine the T_c_ (°C), T_m_ (°C), and ΔH (J/g) of the cryopreservation media [82]. T_c_ is not a thermodynamic property but the nucleation temperature in the sample under the experimental conditions. T_m_ is the temperature under which crystallization can occur. ΔH corresponds to the amount of energy which is released during ice crystallization on cooling and allows the quantification of crystallized ice in the solution. T_m_ was measured during a period of rapid cooling (100 °C/min) followed by a period of slow warming (2.5 °C/min) and was defined at the top of the main melting peak. T_c_ and ΔH were determined using a period of slow cooling (2.5 °C/min) followed by a period of rapid warming (20 °C/min). T_c_ was determined at the onset of the crystallization peak, while ΔH was measured by evaluating the area encompassed between the peak of crystallization and the baseline and calculated using a sigmoid curve baseline.

### 4.9. Statistical Analysis

The statistical analyses of tissue freezing conditions and tissue-derived rbF qualities were performed using GraphPad Prism software V8.0 (GraphPad Inc., La Jolla, CA, USA). A one-way ANOVA and Tukey’s adjustment were used for multiple comparisons of qualitative variables (e.g., tissues, treatments, piece sizes, media, and DMSO concentrations). The unpaired T test was used for simple comparisons between fresh and frozen tissues at a fixed error level of 5%. The statistical analyses of PSC freezing conditions were performed using R software V2.14.1 (R Development Core Team, Vienna, Austria). The results represent three replicates per condition and are presented as mean values ± standard deviations. The cell viability, growth recovery, and gene expression were analyzed using a linear model in which the medium, DMSO concentration and freezing method were included as fixed effects. To determine the best fitting model, the Aikaike Information Criterion for small samples was applied [83,84]. All the conditions were compared to the control treatment: Freezing container, FBS (mESC and rbESC) or KOSR (rbiPSC) medium, and 10% DMSO. The Tukey adjustment was used to compare each condition at a fixed error level of 5%. Differences with *p* values < 0.05 were considered statistically significant. The thermodynamic values were analyzed using descriptive statistics.

## 5. Conclusions

Our study has led to the development of two freezing protocols that will be useful for preserving biodiversity and have been made available on the CRB-Anim website [3]. The first protocol involves the cryopreservation of ear biopsies that will later be thawed and used to derive rbFs that can be reprogrammed into rbiPSCs. The ear biopsies can be processed after storage in PBS-PSG at 4 °C for up to 48 h and are then dissected to separate the skin and cartilage. The minced tissues are frozen to −80 °C in CRYO3 containing 4% DMSO using a controlled-rate freezer at a temperature decrease of 1 °C/min. The second protocol defines the rbPSC freezing conditions that maximize the viability and qualities of the cells after recovery. The dissociated cells are resuspended in culture medium and an equivalent volume of CRYO3 containing 8% DMSO and frozen in a freezing container at −80 °C for 15 h before storage in liquid nitrogen. 

## Figures and Tables

**Figure 1 ijms-21-07285-f001:**
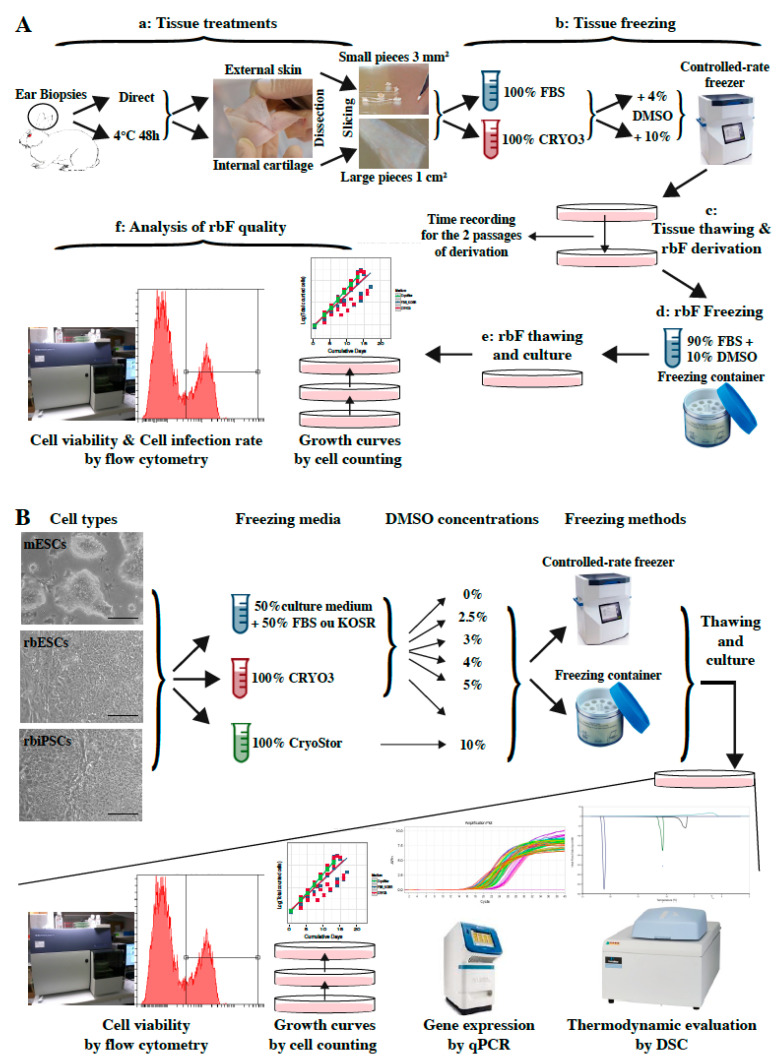
Designs of two experimental cryopreservation studies. (**A**) Analysis of the somatic tissue cryopreservation conditions required to obtain reprogrammable rabbit fibroblasts (rbFs) in rabbit pluripotent stem cells (rbiPSCs). (**B**) Evaluation of the optimal freezing conditions for rabbit pluripotent stem cells (rbPSCs) versus mouse embryonic stem cells (mESCs). Scale bar = 100 μm.

**Figure 2 ijms-21-07285-f002:**
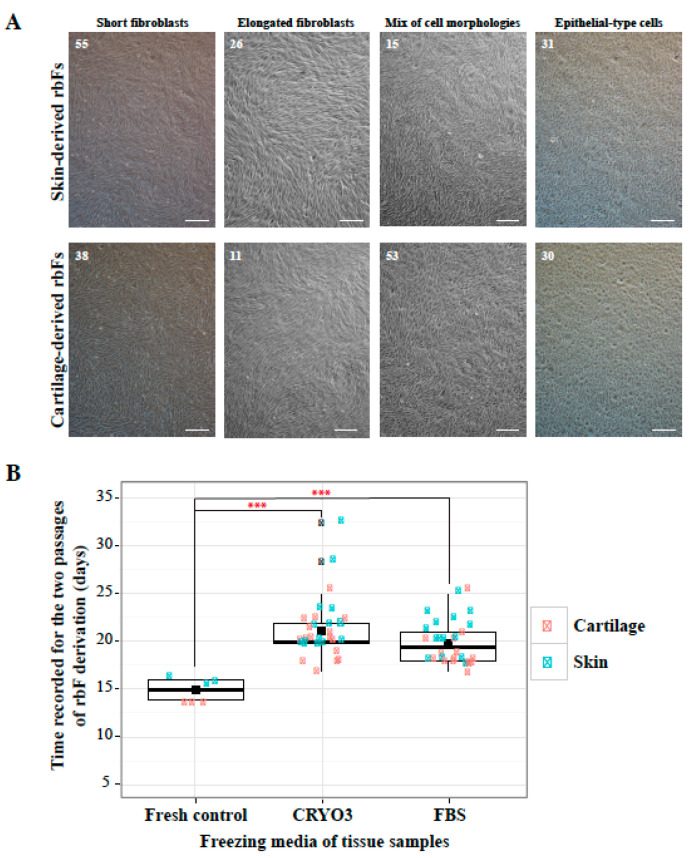
Derivation of rabbit fibroblast (rbF) lines from frozen tissues. (**A**) Bright light microscopic images depicting four morphologic types of cells derived from frozen skin or cartilage. The numbers in the upper left corners correspond to the numbers of the rbF lines (see Table 1 and Appendix A). Scale bar = 50 μm. (**B**) Box plot comparison of the times of rbF derivation during the two passages before freezing between cells issued from fresh tissue, frozen skin, and frozen cartilage. The mean values are indicated by black squares. The black error bars correspond to samples considered to be outliers according to descriptive statistics (GraphPad Prism8 software). The derivation speeds of frozen and fresh tissue-derived rbFs differed significantly (*** unpaired T test, *p* < 0.0001).

**Figure 3 ijms-21-07285-f003:**
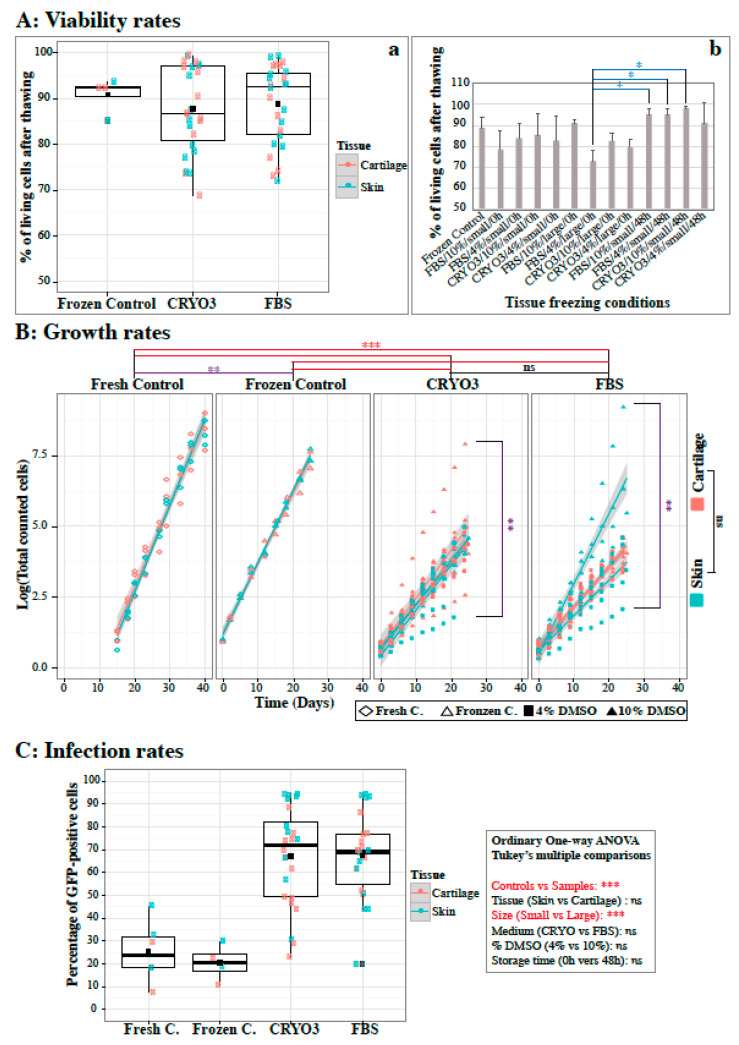
Effects of tissue freezing conditions on the qualities of derived, frozen and thawed rabbit fibroblasts (rbFs). (**A**) Box plot and histogram comparisons of the rbF viability rates at thawing. (**Aa**) Cells issued from fresh and frozen tissues in CRYO3- or FBS-based media were compared. (**Ab**) Viability of rbFs derived from tissues frozen in 12 different conditions were shown. (**B**) Growth curves of rbFs during six passages after thawing. Freshly derived cells were compared with frozen cells issued from fresh and frozen tissues in CRYO3- or FBS-based media. (**C**) Box plot comparison of the GFP-expressing Sendai virus infection rates in rbFs at passage 4 after thawing. Freshly derived cells were compared with frozen cells issued from fresh and frozen tissues in CRYO3- or FBS-based media. Ordinary one-way ANOVA: Tukey’s multiple comparisons: *** *p* < 0.001 (in red); ** *p* < 0.01 (in purple); * *p* < 0.05 (in blue); ns, not significant (in black).

**Figure 4 ijms-21-07285-f004:**
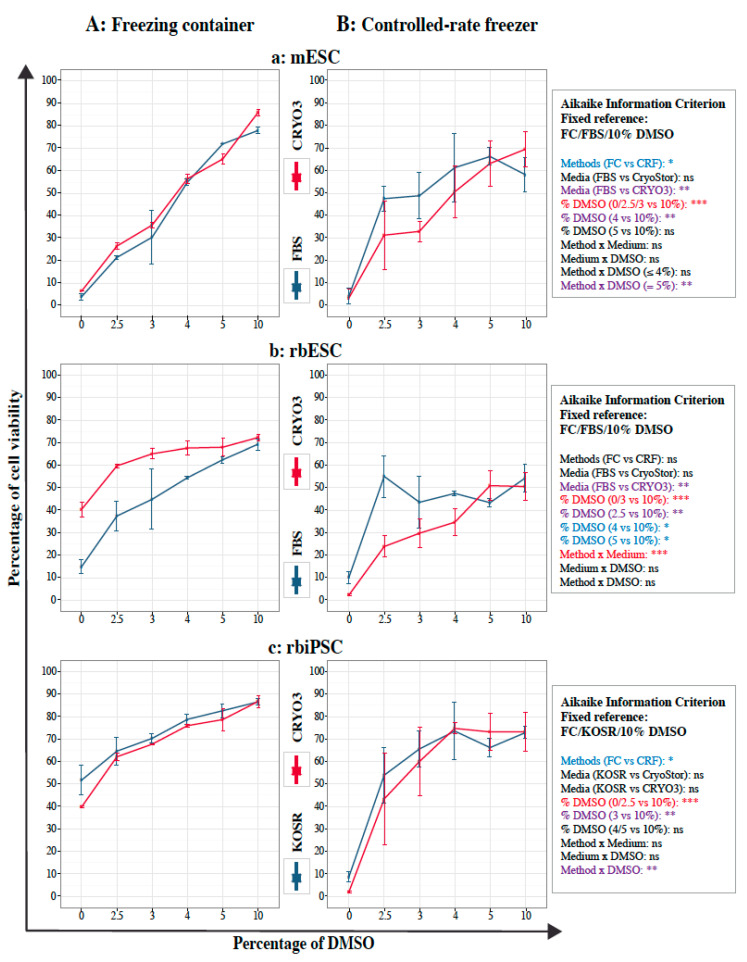
Viability rates of pluripotent stem cells (PSCs) after thawing according to the percentage of DMSO. (**A**) Cells frozen in a freezing container (FC); (**B**) Cells frozen in a controlled-rate freezer (RFC); (**a**) mouse embryonic stem cells (mESCs) frozen in FBS- or CRYO3-based media; (**b**) rabbit embryonic stem cells (rbESCs) frozen in FBS- or CRYO3-based media; (**c**) rabbit induced pluripotent stem cells (rbiPSCs) frozen in KOSR- or CRYO3-based media. Before freezing, the viability rates were 92.33 ± 2.45%, 88.17 ± 1.52%, and 92.93 ± 1.27% for mESCs, rbESCs, and rbiPSCs, respectively. After thawing, the respective viability rates of the positive controls, cells frozen in CryoStor^®^ CS10 (Stemcell Technologies, Grenoble, France) in a freezing container and controlled-rate freezer, were 86.43 ± 1.59% and 84.53 ± 0.58% for mESCs, 71.07 ± 2.08% and 8.20 ± 1.50% for rbESCs, and 89.60 ± 0.17% and 84.80 ± 2.56% for rbiPSCs. The best fitting model was defined by the Aikaike Information Criterion for small samples and the Tukey adjustment were used for comparison between each condition and the fixed condition usually used for each type of PSCs: *** *p* < 0.001 (in red); ** *p* < 0.01 (in purple); * *p* < 0.05 (in blue); ns, not significant(in black).

**Figure 5 ijms-21-07285-f005:**
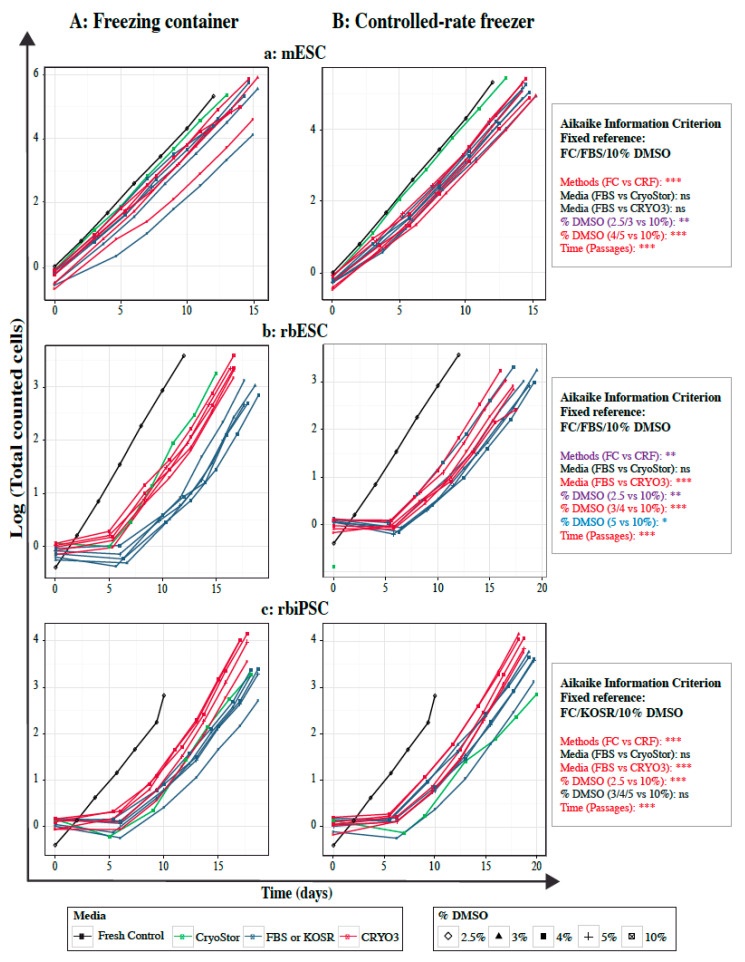
Average growth curves of three samples per cultured pluripotent stem cell (PSC) line during six passages after thawing. Each diagram shows the growth curves of cells frozen in CRYO3- (red) and FBS/KOSR-based (blue) media according to the percentage of DMSO and of the corresponding cell controls (freshly derived cells (black) and cells frozen in CryoStor^®^ CS10 (green)). (**A**) Cells frozen in a freezing container (FC); (**B**) Cells frozen in a rate-controlled freezer (CRF); (**a**) mouse embryonic stem cells (mESCs); (**b**) rabbit embryonic stem cells (rbESCs); (**c**) rabbit induced pluripotent stem cells (rbiPSCs). The best fitting model was defined by the Aikaike Information Criterion for small samples and the Tukey adjustment were used for comparison between each condition and the fixed condition usually used for each type of PSCs: *** *p* < 0.001 (in red); ** *p* < 0.01 (in purple); * *p* < 0.05 (in blue); *ns*, not significant (in black).

**Figure 6 ijms-21-07285-f006:**
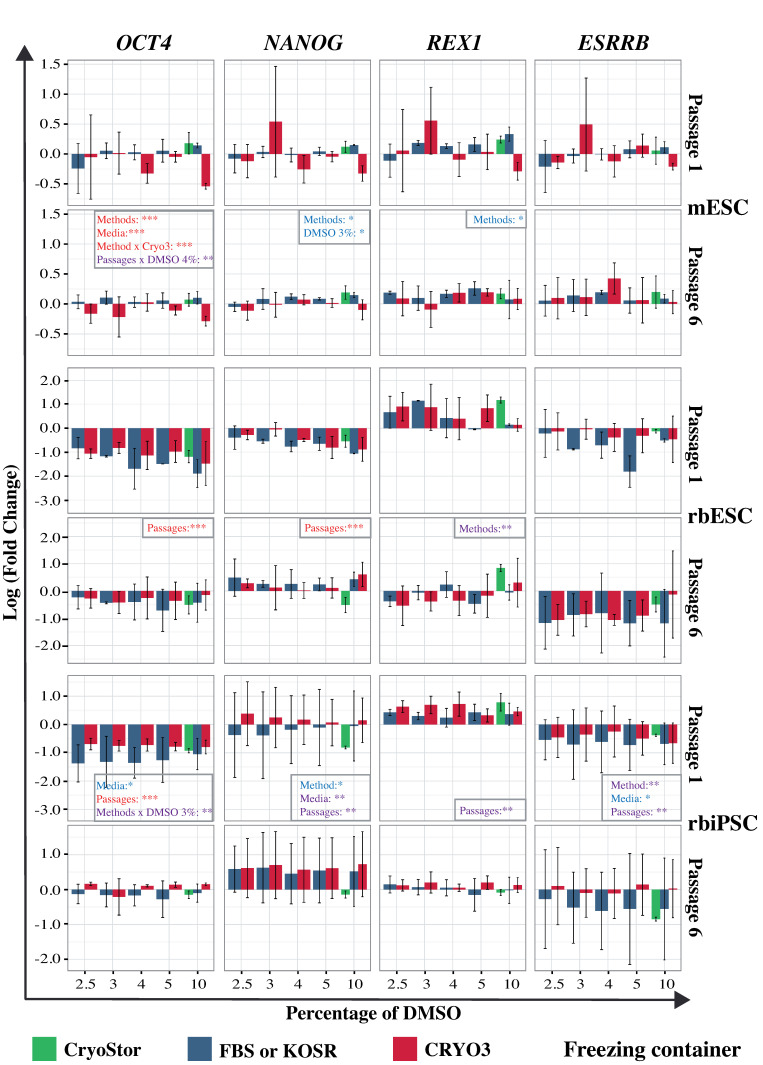
Gene expression in pluripotent stem cells (PSCs) frozen in a freezing container (FC) according to the freezing media and the concentration of DMSO. The expression of the pluripotency genes *OCT4*, *NANOG*, *REX1*, and *ESRRB* was analyzed in cells cultured during passages 1 and 6 after thawing. The graphs depict the logarithmic values of fold changes, which were normalized to rabbit *TBP* or mouse *Actb* expression and standardized to the mean values of fresh samples (three independent replicates) for each condition. The best fitting model was defined by the Aikaike Information Criterion for small samples and the Tukey adjustment were used for comparison between each condition and the fixed condition usually used for each type of PSCs: FC/FBS/10%/P1 for mouse (mESCs) and rabbit (rbESCs) embryonic stem cells and FC/KOSR/10%/P1 for rabbit induced pluripotent stem cells (rbiPSCs). Only significant differences between all studied parameters are indicated: *** *p* < 0.001 (in red); ** *p* < 0.01 (in purple); * *p* < 0.05 (in blue).

**Table 1 ijms-21-07285-t001:** Treatments of ear biopsies and descriptions of derived rbF lines.

Biopsy Treatments	Piece Sizes	Type of Tissues	Tissue Freezing		Quantity of Biopsies	Numbers of Derived rbF Lines	Freezing of rbF Lines at P2 *	Types of rbF Lines
			Media	% DMSO				
Direct	3 mm^2^	Skin	/	/	3	1; 3; 5	N	rbF fresh controls
		Cartilage	/	/	3	2; 4; 6	N	
		Skin	/	/	2	71; 73	Y	rbF frozen controls
		Cartilage	/	/	2	72; 74	Y	
		Skin	FBS	10	4	7; 15; 55; 63	Y	Frozen tissue-derived rbFs
				4	4	8; 16; 56; 64	Y	
			CRYO3	10	4	9; 17; 57; 65	Y	
				4	4	10; 18; 58; 66	Y	
		Cartilage	FBS	10	4	11; 19; X; 67	Y	
				4	4	12; 20; 60; 68	Y	
			CRYO3	10	4	13; 21; 61; 69	Y	
				4	4	14; 22; 62; 70	Y	
	1 cm^2^	Skin	FBS	10	2	39; X	Y	
				4	2	40; 48	Y	
			CRYO3	10	2	41; 49	Y	
				4	2	42; 50	Y	
		Cartilage	FBS	10	2	43; 51	Y	
				4	2	44; 52	Y	
			CRYO3	10	2	45; 53	Y	
				4	2	46; 54	Y	
4 °C/48 h	3 mm^2^	Skin	FBS	10	2	23; 31	Y	
				4	2	24; 32	Y	
			CRYO3	10	2	25; 33	Y	
				4	2	26; X	Y	
		Cartilage	FBS	10	2	27; 35	Y	
				4	2	28; 36	Y	
			CRYO3	10	2	29; 37	Y	
				4	2	30; 38	Y	
	1 cm^2^	Skin	FBS	10	2	X	/	No rbF derivation from frozen tissues
				4	2	X	/	
			CRYO3	10	2	X	/	
				4	2	X	/	
		Cartilage	FBS	10	2	X	/	
				4	2	X	/	
			CRYO3	10	2	X	/	
				4	2	X	/	

Each biopsy per treatment and freezing condition was derived from a different rabbit; the ears of 11 rabbit females were used. / = non relevant; X = No derivation of rbFs from thawing tissue samples. * P2 = Passage 2; N = No freezing and Y = Freezing; rbF = rabbit fibroblast; FBS = fetal bovine serum.

**Table 2 ijms-21-07285-t002:** Thermodynamic properties of the freezing media.

Cell Types	Media	DMSO Concentrations	T_c_ (°C) *	T_m_ (°C) **	∆H (J/g) ***
mESCs	FBS	0%	−16.73 ± 1.01	0.99 ± 0.04	−280.27 ± 4.01
		5%	−15.97 ± 4.33	−1.56 ± 0.08	−224.78 ± 4.31
		10%	−18.68 ± 1.81	−3.61 ± 0.34	−185.06 ± 3.11
rbESCs	FBS	0%	−17.95 ± 1.25	0.71 ± 0.28	−274.05 ± 5.10
		5%	−19.65 ± 1.01	−2.53 ± 0.42	−205.53 ± 0.98
		10%	−19.19 ± 4.23	−3.82 ± 0.12	−184.05 ± 3.11
rbiPSCs	KOSR	0%	−18.22 ± 4.58	0.45 ± 0.06	−256.32 ± 5.51
		5%	−21.74 ± 3.35	−1.70 ± 0.27	−215.06 ± 1.20
		10%	−19.50 ± 2.23	−3.99 ± 0.14	−175.71 ± 0.11
All cell types	CRYO3	0%	−16.61 ± 1.96	0.51 ± 0.10	−263.15 ± 1.32
		5%	−15.66 ± 2.83	−1.69 ± 0.28	−224.34 ± 9.40
		10%	−21.32 ± 3.35	−4.20 ± 0.16	−183.70 ± 7.28
All cell types	CryoStor	10%	−22.42 ± 1.63	−4.45 ± 0.35	−158.61 ± 1.33

Results are presented as the means ± standard deviations of three replicates per medium type. * T_c_: Crystallization temperature; ** T_m_: Melting temperature; *** ∆H: Maximum crystallization enthalpy.

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
