# Peer review of "Insights into Species Preservation: Cryobanking of Rabbit Somatic and Pluripotent Stem Cells"

_ijms, 2020, doi:10.3390/ijms21197285_

Round 1

Reviewer 1 Report

The manuscript:  “Insights into species preservation: cryobanking of rabbit somatic and pluripotent stem cells” by Gavin-Plagne et al. aims at 1). Defining the cryopreservation conditions required to obtain reprogrammable rbFs in rbiPSCs from somatic tissues; and 2).  Determining the appropriate freezing conditions for rbPSCs versus mESCs (line 108).

To achieve these goals, 2 sets of experiments were designed and performed. First experiment is to isolate the rabbit ear and freeze down the tissue under various conditions. Then these tissues were thawed and the time required for the cells to reach confluency; cell viability, growth curve and the infection rate were compared as readout.  The factors involved in this experiment are numerous, which are difficult to compare statistically. The author failed to give clear conclusion that which is the optimal cryopreservation condition for obtaining reprogrammable rbFs. Most importantly, is the cryopreservation condition really the major factor to affect the reprogrammability of the rbFs?

The second experiment designed to compare the different cryopreservation condition in the viability, growth curve and gene expression profile of the mESCs, rbESC and rbiPSC. Again, the authors showed panels of data with various conditions, but the conclusion drawn from these data lacks statistical significant, therefore is not very convincing.

Overall, the rational of the work needs to be clarified. And the data presentation needs to be improved by incorporating statistical analysis.  

Author Response

We would like to thank the reviewer for his comments, which helped us to improve our manuscript.

Reviewer 1

Comment #1:

To achieve these goals, 2 sets of experiments were designed and performed. First experiment is to isolate the rabbit ear and freeze down the tissue under various conditions. Then these tissues were thawed and the time required for the cells to reach confluency; cell viability, growth curve and the infection rate were compared as readout. The factors involved in this experiment are numerous, which are difficult to compare statistically. The author failed to give clear conclusion that which is the optimal cryopreservation condition for obtaining reprogrammable rbFs.

Following the reviewer’s comment, we realized that our results were not sufficiently explained. We have detailed them by adding several sentences in the results, a histogram with statistical analysis in the figure 3A, statistical data in figure 3B and 3C and a conclusion about the optimal cryopreservation condition of tissue. We have also modified the discussion section accordingly. All these inclusions are indicated in red in the revised manuscript.

In the results (lines 203-213):

“Immediately after thawing, the viability rates of rbF lines ranged from 68.7% to 99.2% and no difference was observed between skin- (in green) or cartilage- (in pink) derived rbF lines for both samples and control (Figures 3Aa and S2A). We therefore compared the viability of rbF derived from the 12 freezing conditions used (Figure 3Ab); only four of them showed significant differences (rbF derived from tissue frozen in FBS/10%/large/0h versus those derived from tissue frozen in FBS/10%/small/48h, FBS/4%/small/48h or CRYO3/10%/small/48h). According to the composition of these four freezing conditions, the common parameter that seems to be responsible for this difference is the size of the frozen fragments. Nevertheless, very few samples were included in these four freezing conditions, which may bias the statistical test. Viability rates were more likely to be directly related to the qualities of the rbF freeze-thaw procedure rather than to the freezing conditions of the original tissues.”

In the results (lines 216-217, Lines 221-223 and lines 224-226):

No statistical difference in rbF growth were induced by the original medium or tissue used for freezing.”

“Similarly, the mean slope of the fresh rbF controls was statistically higher than that of frozen rbF controls (p <0.01), again showing an effect of the freezing procedure on cell growth.”

“The infection rates of rbFs from frozen tissues ranged from 19.9% to 94.4% (Figure 3), and no significant correlations were observed between the infection rate and the tissue origin or freezing condition, except for the size of tissue fragments (Figure S2C).”

The conclusion of this results section (lines 237-242):

“Overall, these data showed that the percentage of DMSO and the size of tissue fragments have a substantial effect on the quality of frozen tissue-derived rbFs. The original tissue, the medium and the storage time of the tissue before freezing have no significant impact on any of the analyzed factors. Finally, the best conditions for freezing ear tissue to obtain reprogrammable rbFs are treatment of small pieces of skin or cartilage, with or without prior storage of the ears for 48h in PBS at 4°C, in a medium based on CRYO3 or FBS with 10% DMSO, using a controlled-rate freezer.”

Figure 3A, 3B and 3C + lines 245-247 + lines 251-253:

“(A) Box plot and histogram comparisons of the rbF viability rates at thawing. (Aa) Cells issued from fresh and frozen tissues in CRYO3- or FBS-based media were compared. (Ab) Viability of rbFs derived from tissues frozen in 12 different conditions were shown.”

“Ordinary One-Way ANOVA: Tukey’s multiple comparisons: *** p < 0.001 (in red); ** p < 0.01 (in purple); * p < 0.05 (in blue); ns no significant (in black).”

In the discussion (lines 455-459):

“Only two conditions, the size of the frozen fragments and the percentage of DMSO, influenced the viability and transduction rates of the rbF lines and the growth of thawed cells, respectively. However, it should be noticed that the low number of samples included in some analyzed groups may have biased the statistical tests.”

Comment #2:

Most importantly, is the cryopreservation condition really the major factor to affect the reprogrammability of the rbFs?

We agree with the reviewer that the cryopreservation condition is probably not the main factor affecting rbFs reprogramming. Nevertheless, it is one of the factors to consider, including chromatin status, epigenome configuration and accumulation of mutations in the genome that is related to the age of the tissue donor. Since it has been shown that the freezing procedure alters the epigenome of the cells and consequently the fate of the frozen cells, the quality of the freezing procedures must be taken into account. We have added sentences and four references (#26-29) in the introduction to clarify this point.

In the introduction (lines 78-81, in red):

“Culture but also freezing can induce the selection of sub-populations of PSCs presenting mutations [26] or epigenetic modifications with long-term putative effects on cells and/or their derivatives [27, 28]. Similarly, somatic reprogramming is regulated by epigenetic phenomena [29]  that could be affected by the epigenetic status of the cells to be reprogrammed.”

Comment #3:

The second experiment designed to compare the different cryopreservation condition in the viability, growth curve and gene expression profile of the mESCs, rbESC and rbiPSC. Again, the authors showed panels of data with various conditions, but the conclusion drawn from these data lacks statistical significant, therefore is not very convincing.

As for comment #1, we have revised sections 2.4-2.6 and added statistics in the figures. These inclusions are indicated in red in the revised manuscript. The blue texts of sections 2.4 and 2.5 correspond to answers to reviewer #2.

Section 2.4 (Viability of frozen-thawed PSCs) (Lines 255-300):

“Before freezing, the viability rates of mESCs, rbESCs and rbiPSCs were 92.33 ± 2.45%, 88.17 ± 1.52% and 92.93 ± 1.27%, respectively. After thawing, the viability rates of all three cell types decreased significantly as the percentage of DMSO decreased, particularly when the percentage of DMSO was 4% or less, regardless whether FBS- or CRYO3-based freezing media had been used (Figure 4). For mESCs and rbiPSCs, significant differences in viability were also observed between cells frozen in a freezing container (Figure 4A) and the controlled-rate freezer (Figure 4B); the former was associated with less variability and better viability regardless of the concentration of DMSO. Larger error bars were observed for the controlled-rate freezer (Figure 4B) compared with the freezing container (Figure 4A). A difference in the cooling curves used in the two methods might explain these larger error bars. The cooling curve was linear in the controlled-rate freezer while the cooling curve slowed down considerably at the end of the cooling cycle with the freezing container. The cooling curve of the controlled-rate freezer might not be appropriate or not similar enough to that of the freezing container in the -80°C freezer. A second explanation is that the cell lines used for this study are usually frozen in a freezing container and the cells are somehow adapted to this method. On the contrary, the cells are not adapted to freezing by the controlled-rate freezer method which would then induce more cellular stress.

Similar rates of viability were observed for mESCs frozen in media containing 10% DMSO, CryoStor® CS10 (86.43 ± 1.59%) and fresh cells (92.33 ± 2.45%) when a freezing container was used (Figure 4Aa). Lower viability rates were achieved with a controlled-rate freezer than with a freezing container under most conditions, even for positive control cells (cells frozen in CryoStor® CS10; 84.53 ± 0.58%) (Figure 4Ba). However, when using the controlled-rate freezer, no significant difference was observed for mESCs frozen in FBS- or CRYO3-based media containing 5% or 10% DMSO (Figure 4Ba). The interaction between the method and the percentage of DMSO was only significant at 5% of DMSO.

Most of the parameters had a significant effect on the viability of rbESCs and the interaction between the method and the medium was noteworthy (Figure 4b): CRYO3-based medium improved the viability of rbESCs frozen in a freezing container, while FBS-based medium improved the viability when a controlled-rate freezer was used. However, the viabilities observed with a freezing container using 10% DMSO were similar between the positive control (cells frozen in CryoStor® CS10; 71.07 ± 2.08%) and samples frozen in both media (Figure 4Ab) and lower than those of fresh cells (88.17 ± 1.52%), indicating sub-optimal freezing conditions for the rbESCs, even with the freezing container.

No significant difference was observed between rbiPSCs frozen in CRYO3- or FBS-based media containing 4%, 5% and 10% DMSO with both methods (Figure 4c). In addition, the viability rates of the latter conditions did not significantly differ from those of fresh sample (92.93 ± 1.27%) and positive control (cells frozen in CryoStor® CS10; 89.60 ± 0.17%) when using a freezing container (Figure 4Ac). The interaction between the method and the percentage of DMSO was also highly significant, confirming the sub-optimal freezing conditions used with the controller-rate freezer.

Overall, these results are consistent with the ability of both synthetic media, CRYO3 and CryoStor CS10, to preserve the viability of mouse and rabbit PSCs after a freezing-thawing cycle in a freezer container. In addition, the level of DMSO in the CRYO3-based medium can be reduced without affecting cell viability to 5 or 4%. These data also show that the decreasing temperature curve must be better suited to efficiently use a rate-controlled freezer. Nevertheless, after five cell culture passages, the viability rates of all cell types were similar to those of the fresh cells, regardless the initial freezing conditions (Figure S3), showing no effect of freezing conditions on the viability of recovered cell.”

Figure 4 (Lines 309-312) and Figure 5 (Lines 356-359):

“The best fitting model was defined by the Aikaike Information Criterion for small samples and the Tukey adjustment were used for comparison between each condition and the fixed condition usually used for each type of PSCs: *** p < 0.001 (in red); ** p < 0.01 (in purple); * p < 0.05 (in blue); ns no significant (in black).”

Section 2.5 (Growth curves of frozen-thawed PSCs) (Lines 314-349):

“Compared to fresh cells, frozen rabbit PSCs required a recovery period of at least 5 days before reaching the normal growth rate, regardless of the freezing medium, freezing method and DMSO concentration. This recovery period was also observed with the control cells frozen in the CryoStor CS10 (Figures 5b and 5c). In addition, under certain freezing conditions (in FBS/KOSR-based media and CryoStor®CS10), rbPSC growth curves showed an inflection during the first five days of culture, which is possibly due to a higher rate of cell death, resulting in a longer recovery period. In contrast, frozen mouse cells recovered rapidly and grew at their usual rate (Figure 5a). This difference in recovery between mouse and rabbit PSCs may be due to the intrinsic stability, better resistance to cell dissociation and higher viability of mPSCs relative to rbPSCs.

 Only time, freezing method and DMSO concentration had significant effects on the total numbers of recovered mESCs (Figure 5a). No significant difference was noticed between curve slopes of fresh cells, control cells and sample cells frozen in FBS- or CRYO3-based media in a freezing container, particularly with high concentration of DMSO (Figure 5Aa and S4). In the same way, similar curve slopes were observed between mESCs frozen in FBS- or CRYO3-based media in a controlled-rate freezer, regardless the concentration of DMSO (Figure 5Aa and S3). However, the latter conditions resulted in slower growth of the frozen mESCs than that of fresh or control cells.

­For rbPSCs, all the fixed parameters (e.g., freezing method, freezing medium and DMSO concentration), as well as time, significantly influenced the total numbers of cells recovered (Figures 5b and 5c). Irrespective of method and DMSO concentration, CRYO3-based medium showed significantly increased growth of rbiPSCs and rbESCs than FBS/KOSR-based medium and CryoStor CS10 control medium. In addition, no significant difference was observed in the total rbiPSCs recovered after freezing in 3, 4, 5 and 10% of DMSO (Figure 5C and S4), regardless of the medium and method used, suggesting that rabbit cells could be cryopreserved with low concentration of cryoprotectant. The same trend seems to be true for the rbESCs frozen with 5% of DMSO (Figure 5b and S4), but should be confirmed with more optimal freezing conditions, particularly with the controlled rate freezer. Once rabbit cells had fully recovered, the slopes of the linear portion of the growth curves were similar within each cell type frozen in a defined medium, regardless of the concentration of DMSO (Figure S4). However, as previously described for rabbit cells, the CRYO3-based medium induced less cell death and allowed faster recovery during the first few days of culture (Figure 5B and 5c). In addition, compared to FBS/KOSR-based media, a lower fluctuation in cell growth rates between replicates was observed with CRYO3-based medium (Figure S4).

In conclusion, the relative fragility of rbESCs compared to mESCs has influenced the recovery and growth of the frozen cells under all freezing conditions, indicating the need to improve their cryopreservation methods. However, the CRYO3-based medium and the low percentage of cryoprotectant (4-5% of DMSO) appeared to increase the growth rate of rabbit cells compared to the FBS/KOSR-based medium usually used with 10% DMSO.”

Section 2.6 (Gene expression in frozen-thawed PSCs) (Lines 374-396):

For mESCs, we mainly observed significant effects of the freezing method on expression of Oct4, Nanog and Rex1 genes. The linear model used did not reveal any significant difference in Essrb expression (Figure 6). For both rbESCs and rbiPSCs, the main factor showing a significant effect on the expression of most of the six genes analyzed (OCT4, NANOG, REX1, ESRRB, CDH1 and CDH2) was the passages. The freezing medium and method also induced a significant individual effect on the expression of all five genes, with the exception of REX1, in rbiPSCs. In contrast, the expression of REX1, like that of CDH2, was significantly influenced by the freezing method in rbESCs. In the latter, the freezing medium induced significant variations in the expression of CDH1 and CDH2. Surprisingly, the percentage of DMSO itself did not generate significant variations in gene expression, with the exception of Nanog in mESCs and CDH2 in rbESCs. However, we observed that at passage 6, the gene expression patterns of samples frozen with 3, 4 or 5% DMSO were generally more similar to those of fresh samples than to those of samples frozen with other DMSO concentrations, especially with the traditional 10% concentration. More importantly, gene expressions obtained with control cells frozen in CryoStor CS10 were not significantly closer to those of fresh cells and in some cases even lower than those of rbPSCS frozen in FBS/KOSR- and CRYO3-based media containing 10% DMSO.

Overall, our data showed that PSC freezing conditions influenced the expression of pluripotency network and marker genes after a few days in culture. The more the freezing conditions were adapted to the cells, the closer the expression levels were to those of fresh cells, as shown by the mESC expression results. However, after the recovery of the cells for five passages in culture, gene expressions tented to stabilize at the same levels than those of fresh cells, especially in cells frozen with 3-5% DMSO in CRYO3-based medium using a freezing container or in FBS/KOSR-based medium using a controlled-rate freezer.”

Figure 6 (Lines 403-407):

“The best fitting model was defined by the Aikaike Information Criterion for small samples and the Tukey adjustment were used for comparison between each condition and the fixed condition usually used for each type of PSCs: FC/FBS/10%/P1 for mESCs and rbESCs and FC/KOSR/10%/P1 pour rbiPSCs. Only significant differences between all studied parameters are indicated: *** p < 0.001 (in red); ** p < 0.01 (in purple); * p < 0.05 (in blue).”

In the materials and methods (section 4.9. Statistical analysis) (Lines 680-681):

“All the conditions were compared to the control treatment: Freezing container, FBS (mESC and rbESC) or KOSR (rbiPSC) medium and 10% DMSO.”

Comment #4:

Overall, the rational of the work needs to be clarified. And the data presentation needs to be improved by incorporating statistical analysis.

In order to clarify the manuscript, we have added a few sentences and to improve the presentation of the data, we have provided p-values in the figures. Some of these additions have already been described in response to comments 1, 2 and 3. All other inclusions are marked in red as follows:

In the introduction (lines 78-81):

“Culture but also freezing can induce the selection of sub-populations of PSCs presenting mutations [26] or epigenetic modifications with long-term putative effects on cells and/or their derivatives [27, 28]. In the same way, somatic reprogramming is regulated by epigenetic phenomena [29]  that could be affected by the epigenetic state of the cells to be reprogrammed.”

In the results (lines 116-126):

“Regarding biodiversity preservation, useful somatic tissues must be readily available for sampling, even from field-raised animals and sick or dead individuals, and for cryopreservation without the need for prior cell isolation. Accordingly, a maximum number of biopsy cells should remain alive and be easily derivable after the tissue is frozen and thawed, and the process should not require the complex vitrification procedures needed to preserve tissue structures and vascularization [48, 49]. Therefore, we selected ear biopsies that met these requirements and determined the ideal slow-freezing cryopreservation conditions by testing five parameters: (i) the eventual storage of samples at 4°C for 48 h before cryobanking, (ii) the tissues dissected from the biopsies (skin versus internal cartilage), (iii) the sizes of the frozen tissue pieces (3 mm2 versus 1 cm2), (iv) the freezing medium [synthetic (CRYO3) versus biological (FBS)] and (v) the percentage of the cryoprotective agent (DMSO; 4% versus 10%).”

In the results (lines 148-154):

“Both rbPSCs and mESCs are generally frozen under standard conditions (i.e., 90% FBS or knockout serum replacement (KOSR) + 10% DMSO in a freezing container). However, we observed that the cell mortality at thawing and cell passages is much higher for rbPSCs than for mESCs. This difference is related to the intrinsic mechanism supporting pluripotency in these two species [24], and consequently to the lower resistance of rbPSCs to single-cell dissociation compared to mESCs. In order to improve the freezing conditions of rbPSCs, we analyzed three parameters: the freezing medium, the percentage of cryoprotectant, and the slow-freezing method.”

Figure 2 + line 195: (*** unpaired T test, p<0.0001)

In the discussion (lines 434-439):

“With respect to biodiversity preservation, as described above, we selected ear biopsies for their ease of sampling under most breeding conditions and their ability to be frozen with a minimum of processing. We determined the best slow-freezing cryopreservation parameters by comparing five conditions: (i) immediate freezing versus initial storage of biopsies, (ii) skin versus internal cartilage dissected from the biopsies, (iii) small versus large pieces of frozen tissue, (iv) synthetic versus biological freezing medium and (v) low versus high percentage of DMSO.”

Reviewer 2 Report

Comment#1:

“Currently, cells and small tissues are most commonly preserved via controlled-rate cooling in the presence of serum and 10% dimethyl sulfoxide (DMSO) as a cryoprotectant [26] [27]. This technique can be applied easily using a freezing container or controlled-rate freezer and does not require any prior expertise in cryobiology.”

I disagree that the cryopreservation techniques can be easily applicable to cell and tissues. Even for the medium containing serum and DMSO, appropriate cryopreservation studies (including hold time, CRF programs, cell concentration evaluation) will be needed, in particular these studies are important for therapeutic applications. These studies require strong bioprocessing knowledge and expertise.

Comment #2:

Authors need to provide data on the vitality effect or cell survival at least one day after culture as the viability upon thaw may not be a good representative of cryopreservation condition impact. 

Comment #3;

Authors should comment on the relatively large error bars observed for CRF conditions compared with the freezing container

Comment #4:

Line 253-265

“Generally, no significant differences in growth recovery were observed between mESCs, rbESCs and rbiPSCs frozen in CryoStor® CS10 (positive control) and CRYO3- or FBS/KOSR-based media containing 4%, 5% and 10% of DMSO using either freezing method. Moreover, once the cells had fully recovered, the slopes of the linear portions of the growth curves did not differ significantly within each cell type frozen in a defined medium, regardless of the concentration of DMSO used. However, CRYO3-based medium induced a more rapid recovery with less variable cell growth when compared to FBS/KOSR-based media, especially for rbESCs and rbiPSCs (Figures 5b, 5c and S3).”

This results are somewhat contradicting. If there is no significant difference in growth recovery amongst different conditions, why CRYO3-based medium induced a more rapid recovery?

Comment #5:

In the discussion (lines 404 – 408), the authors refer to differences between use of freezing container and CRF but refer to possibility of uncontrolled crystallization as the reasoning. This is surprising as the CRF is supposed to be using a controlled mechanism and freezer container is using an uncontrolled condition. The explanation requires more clarification. In fact, it is possible that the difference observed using CRF may be attributed to suboptimal CRF conditions.

Major comments:

  • The study has provided scientific data around the genetic stability of the cells following different freezing conditions. This is important safety concern that need to be evaluate in short term and long term after the cryopreservation.
  • The study has not provided scientific data around the impact of cryopreservation on the directed differentiation capability of riPSCs. This is one of the defining features of the iPSCs.

Author Response

We would like to thank the reviewer for his comments, which helped us to improve our manuscript.

Reviewer 2

Comment #1:

“Currently, cells and small tissues are most commonly preserved via controlled-rate cooling in the presence of serum and 10% dimethyl sulfoxide (DMSO) as a cryoprotectant [26] [27]. This technique can be applied easily using a freezing container or controlled-rate freezer and does not require any prior expertise in cryobiology.”

I disagree that the cryopreservation techniques can be easily applicable to cell and tissues. Even for the medium containing serum and DMSO, appropriate cryopreservation studies (including hold time, CRF programs, cell concentration evaluation) will be needed, in particular these studies are important for therapeutic applications. These studies require strong bioprocessing knowledge and expertise.

 We fully agree with the reviewer’s comment and realised that our last sentence “does not require any prior expertise” could be misinterpreted. In this sentence, we wished to point out that cell biologists often lack expertise in cryobiology and therefore apply the most commonly used protocols. Most laboratories have already experienced a poor recovery of cells and tissues after freezing-thawing cycles, which is likely to result from inappropriate freezing-thawing protocols. Determining effective biobanking strategies is crucial for every laboratory working with cells and tissues, especially if therapeutic applications are foreseen. We have modified the sentence in order to take the reviewer’s remark into account (lines 86-88 in blue).

This technique can be easily applied using a freezing container or rate-controlled freezer and is most often used by cell biologists lacking expertise in cryobiology.”

Comment #2:

Authors need to provide data on the vitality effect or cell survival at least one day after culture as the viability upon thaw may not be a good representative of cryopreservation condition impact. 

We agree with the reviewer that the effect of cryopreservation conditions on cell viability could mostly be detected after a few days in culture. Unfortunately, we did not analyze membrane integrity after day 0, except for the case of pluripotent stem cells which were retested on the fifth passage (Figure S3). By this time, the cells had already recovered and no effect of cryopreservation on their viability was observed. However, in rbESCs and rbiPSCs, the growth curves obtained by counting cells at each passage clearly show a decrease in the number of cells between day 0 and day 5, as a result of cell death (Figures 5b,c and S4). In contrast, no inflection of the growth curves was observed in mESCs and rbFs (Figure 5a and 3B), suggesting that mESCs and rbFs recovered much faster from the freeze-thaw procedure than rbESCs and rbiPSCs. Therefore, the analysis of membrane integrity at the time of thawing and that of the cell growth during the first few days of culture make it possible to determine the impact of freezing conditions on the cell viability. We have added a sentence about viability at passage 5 (Lines 297-300 in blue) and written a commentary (Lines 317-319 in blue) in the revised manuscript.

“Nevertheless, after five cell culture passages, the viability rates of all cell types were similar to those of the fresh cells, regardless the initial freezing conditions (Figure S3), showing no effect of freezing conditions on the viability of recovered cell.”

“In addition, under certain freezing conditions (in FBS/KOSR-based media and CryoStor®CS10), rbPSC growth curves showed an inflection during the first five days of culture, which is possibly due to a higher rate of cell death, resulting in a longer recovery period.”

Comment #3;

Authors should comment on the relatively large error bars observed for CRF conditions compared with the freezing container

We discussed this point in the revised manuscript (lines 262-270 in blue) as follows:

“Larger error bars were observed for the controller-rate freezer (Figure 4B) compared with the freezing container (Figure 4A). A difference in the cooling curves used in the two methods might explain the larger error bars. The cooling curve was linear in the controller-rate freezer while the cooling curve slowed down considerably at the end of the cooling cycle with the freezing container. The cooling curve of the controller-rate freezer might not be appropriate or not similar enough to that of the freezing container in the -80°C freezer. A second explanation is that the cell lines used for this study are usually frozen in a freezing container and the cells are somehow adapted to this method. On the contrary, the cells are not adapted to freezing by the controller-rate freezer method which would then induce more cellular stress.”

.

Comment #4:

“Generally, no significant differences in growth recovery were observed between mESCs, rbESCs and rbiPSCs frozen in CryoStor® CS10 (positive control) and CRYO3- or FBS/KOSR-based media containing 4%, 5% and 10% of DMSO using either freezing method. Moreover, once the cells had fully recovered, the slopes of the linear portions of the growth curves did not differ significantly within each cell type frozen in a defined medium, regardless of the concentration of DMSO used. However, CRYO3-based medium induced a more rapid recovery with less variable cell growth when compared to FBS/KOSR-based media, especially for rbESCs and rbiPSCs (Figures 5b, 5c and S3).”

These results are somewhat contradicting. If there is no significant difference in growth recovery amongst different conditions, why CRYO3-based medium induced a more rapid recovery?

We agree with the reviewer that our last sentence is unclear. On the one hand, there is no significant difference between the slopes of growth curves with the statistical model used. On the other hand, as described above, the inflections of the curves during the first few days after thawing show variations in the cell death rates and consequently in the recovery times of rabbit cells (Figure 5b, 5c, and S4).  Thus, CRYO3-based medium allows for a faster recovery of rbESCs and rbiPSCs than FBS/KOSR-based media. We have modified the corresponding sentence to hopefully clarify this point (lines 341-344 in blue).

“However, as previously described for rbESCs and rbiPSCs, the CRYO3-based media induced less cell death and allowed a faster recovery during the first few days of culture (Figures 5b and 5c). In addition, compared to FBS/KOSR-based media, a lower fluctuation in cell growth rates between replicates was observed with CRYO3-based media (Figure S4).”

Comment #5:

In the discussion (lines 404 – 408), the authors refer to differences between use of freezing container and CRF but refer to possibility of uncontrolled crystallization as the reasoning. This is surprising as the CRF is supposed to be using a controlled mechanism and freezer container is using an uncontrolled condition. The explanation requires more clarification. In fact, it is possible that the difference observed using CRF may be attributed to suboptimal CRF conditions.

We agree with the reviewer that differences in the observed cell viability between freezing container and CRF might result from sub-optimal CRF conditions. We already discussed this issue in the results section on PSC viability. In this section, we also discussed the possibility of adding a seeding step to improve the freezing protocol. It is well known that this step is essential for embryo freezing. We have modified this paragraph to clarify this point and added reference #73 in the revised manuscript (Line 504-510 in blue).

“As mentioned before, this result can be explained by the previous adaptation of PSCs to the freezing containers or by the suboptimal conditions used with the controlled-rate freezer. In slow-freezing protocols, crystallization control is crucial to avoid the formation of heterogenous and chaotic crystals. This control could be improved by manual nucleation induction or seeding [73]. A seeding step before using the controlled-rate freezer has been shown to improve PSC freezing [30, 47], although important variations were also observed with human PSCs using seeding at -7°C or -10°C [74-76].”

Major comments:

The study has provided scientific data around the genetic stability of the cells following different freezing conditions. This is important safety concern that need to be evaluate in short term and long term after the cryopreservation.

The study has not provided scientific data around the impact of cryopreservation on the directed differentiation capability of riPSCs. This is one of the defining features of the iPSCs.

To our knowledge, the effect of cryopreservation on the differentiation ability of iPSCs has not yet been studied in any species. Our data indicate that, once rbiPSCs have recovered from thawing, their gene expression profiles is similar to that of fresh cells. It is therefore unlikely that freezing/thawing will dramatically alter their differentiation capabilities. In addition, no protocol for iPSC differentiation has yet been validated in rabbits. If rbiPSCs are to be used for species preservation, cell lines should be tested for embryo colonization, gamete differentiation and cloning by nuclear transfer. None of these techniques have been validated in rabbits. We addressed this issue at the end of the discussion in the revised manuscript (Lines 529-532 in blue).

"However, although this technique appears to be successful in preserving the quality of fibroblasts and iPSCs, the genetic and epigenetic heritage of the frozen cells needs to be tested and the differentiation capabilities of the frozen iPSCs need to be verified to ensure complete preservation".

Round 2

Reviewer 1 Report

The authors had addressed all my previous concerns and made significant improvement of the manuscript. The data have now incorporated with statistical analysis wherever possible, and clear conclusions were drawn to each section.

No further concerns about the manuscript.

Reviewer 2 Report

I have reviewed the authors' responses. The modifications they have included in the manuscript are satisfactory and acceptable. I have no further comment.